# Planar cell polarity signaling coordinates oriented cell division and cell rearrangement in clonally expanding growth plate cartilage

Yuwei Li[1], Ang Li[2], Jason Junge[3], Marianne Bronner[1]*

[1]Division of Biology and Biological Engineering, California Institute of Technology, Pasadena, United States; [2]Department of Pathology, University of Southern California, Keck School of Medicine, Los Angeles, United States; [3]Translational Imaging Center, University of Southern California, Los Angeles, United States

**Abstract** Both oriented cell divisions and cell rearrangements are critical for proper embryogenesis and organogenesis. However, little is known about how these two cellular events are integrated. Here we examine the linkage between these processes in chick limb cartilage. By combining retroviral-based multicolor clonal analysis with live imaging, the results show that single chondrocyte precursors can generate both single-column and multi-column clones through oriented division followed by cell rearrangements. Focusing on single column formation, we show that this stereotypical tissue architecture is established by a pivot-like process between sister cells. After mediolateral cell division, N-cadherin is enriched in the post-cleavage furrow; then one cell pivots around the other, resulting in stacking into a column. Perturbation analyses demonstrate that planar cell polarity signaling enables cells to pivot in the direction of limb elongation via this N-cadherin-mediated coupling. Our work provides new insights into the mechanisms generating appropriate tissue architecture of limb skeleton.
DOI: https://doi.org/10.7554/eLife.23279.001

*For correspondence:
mbronner@caltech.edu

## Introduction

A central question in modern biology is how cells build a complex tissue within a four dimensional (xyz and t) context. This is particularly true in developing embryos, in which cells undergo intricate behaviors including proliferation, migration and differentiation, while interacting with similar as well as distinct cell types. Two fundamental cellular processes, oriented cell divisions and cell rearrangements, play important roles during tissue growth (*Morin and Bellaïche, 2011*; *Walck-Shannon and Hardin, 2014*). By orienting the axis of division in a stereotypic direction, oriented cell divisions serve two major purposes: first they can drive body axis elongation, as seen in zebrafish gastrulation (*Gong et al., 2004*); second, they can generate cellular diversity, for example by asymmetrically segregating cell fate determinants to produce one stem cell and one differentiated cell, as observed during Drosophila spermatogenesis (*Yamashita et al., 2003*). Cell rearrangements involve cells exchanging neighbors, which can occur via different mechanisms. One scenario involves junctional remodeling whereby adherens junctions between cells shrink in one direction and extend in the orthogonal direction, as seen in Drosophila epithelia tissues (*Bertet et al., 2004*). A second example is by mediolateral cell intercalation, as exemplified by notochord cells during Xenopus gastrulation (*Wallingford et al., 2000*). Both oriented divisions and cell rearrangements can establish and maintain polarized cell organization that subsequently drives body axis elongation (*Gillies and Cabernard, 2011*; *Lau et al., 2015*). For technical reasons, studies of cellular rearrangement have primarily

focused on either slowly dividing or postmitotic cells (*Gillies and Cabernard, 2011*; *Lau et al., 2015*). Thus, it remains a significant challenge to determine how these cellular behaviors are controlled, coordinated and propagated in proliferative tissues.

The growth plate cartilage of the limb is a good example of such a tissue due to its proliferative capacity and stereotypic architecture (*Kronenberg, 2003*; *Lefebvre and Bhattaram, 2010*). Along the proximodistal axis of the tissue, chondrocytes undergo progressive maturation and collagen deposition; once terminally differentiated, hypertrophic chondrocytes are replaced by osteoblasts that lay down a calcified matrix to form long bones (*Figure 1a*) (*Kronenberg, 2003*). The directional growth of proliferative chondrocytes is critical for normal cartilage homeostasis, morphogenesis and regeneration (*Dodds, 1930*; *Li and Dudley, 2009*; *Le Pabic et al., 2014*). Developing chondrocytes have several hallmark features: they acquire an ellipsoidal shape and orient mitotic figures orthogonal to the tissue proximodistal (PD) axis; clonally related cells are thought to be arranged in columns along the PD axis (*Dodds, 1930*; *Li and Dudley, 2009*; *Le Pabic et al., 2014*). Based on these observations, it has been proposed that chondrocyte column formation is comprised of sequential steps of oriented division and subsequent cell rearrangement (*Figure 1b*) (*Dodds, 1930*).

There are several issues with this simplistic model for chondrocyte column formation. First, the relationship between clones and columns is not clear. Although this simple model predicts that clonally related cells form a single column, some studies suggest that columns may be more complex, with two to three adjacent stacks (*Li and Dudley, 2009*; *Ahrens et al., 2009*). Further complicating the analysis of the embryonic growth plate cartilage, columnar structure cannot be visualized histologically (*Li and Dudley, 2009*). Planar cell polarity (PCP) signaling previously has been suggested in coordinating oriented division and column formation in developing growth plate cartilage (*Li and Dudley, 2009*; *Le Pabic et al., 2014*; *Gao et al., 2011*). However, these studies were based on static analyses, making it difficult to determine whether and how cell rearrangements are affected when PCP signaling activity is abnormal.

To tackle the relationships between oriented division and cell rearrangement in a definitive way, here we develop and apply novel retroviral-based multicolor clonal analysis to the developing limb skeleton of chick embryos. The results show that clonally related cells are arranged in either single columns, or multi-dimensional ones often derived from a single progenitor cell. Live imaging demonstrates that single columns are generated through a pivot-like process that reorients sister cells orthogonal to their original orientation. In contrast, multi-column clones likely form by mediolateral cell interaction, similar to cell behaviors in zebrafish craniofacial cartilage (*Le Pabic et al., 2014*). Focusing on the mechanisms underlying single column formation, we show that after cytokinesis N-cadherin is enriched in the post-cleavage region, enabling sister cell association during their subsequent pivot. We further reveal that PCP signaling couples oriented division with cell pivoting by regulating the expression level of junctional N-cadherin. Taken together, our results suggest that mitosis and cell rearrangement are highly coordinated to control proper tissue architecture of growth plate cartilage.

## Results

### Two types of clones in the proliferative zone: single-column and multi-column clones

Understanding dynamic interactions between clonally related and neighboring unrelated cells requires accurate discrimination of clonal boundaries together with observation of cell behaviors over time. To meet this challenge in limb cartilage growth, we developed a multicolor clonal approach in which replication-incompetent avian (RIA) retrovirus encoding distinct fluorescent proteins were simultaneously infected into chick limbs so that individual clones are marked by distinct colors (*Figure 1c*). Three factors further facilitated the assessment of clonality: first, viral insertion sites in the host chromosomes affected protein expression levels such that even clones with the same color can be distinguished from one another by virtue of fluorescence intensities; second, in some cases, more than one virus infected a single progenitor cell to create a new color that expanded the spectrum of rainbow analysis; third, some identical-colored fluorescent proteins were genetically modified so that that they were targeted to different subcellular localizations. If a chondrocyte column is derived from a single progenitor, this column would be unambiguously monocolor

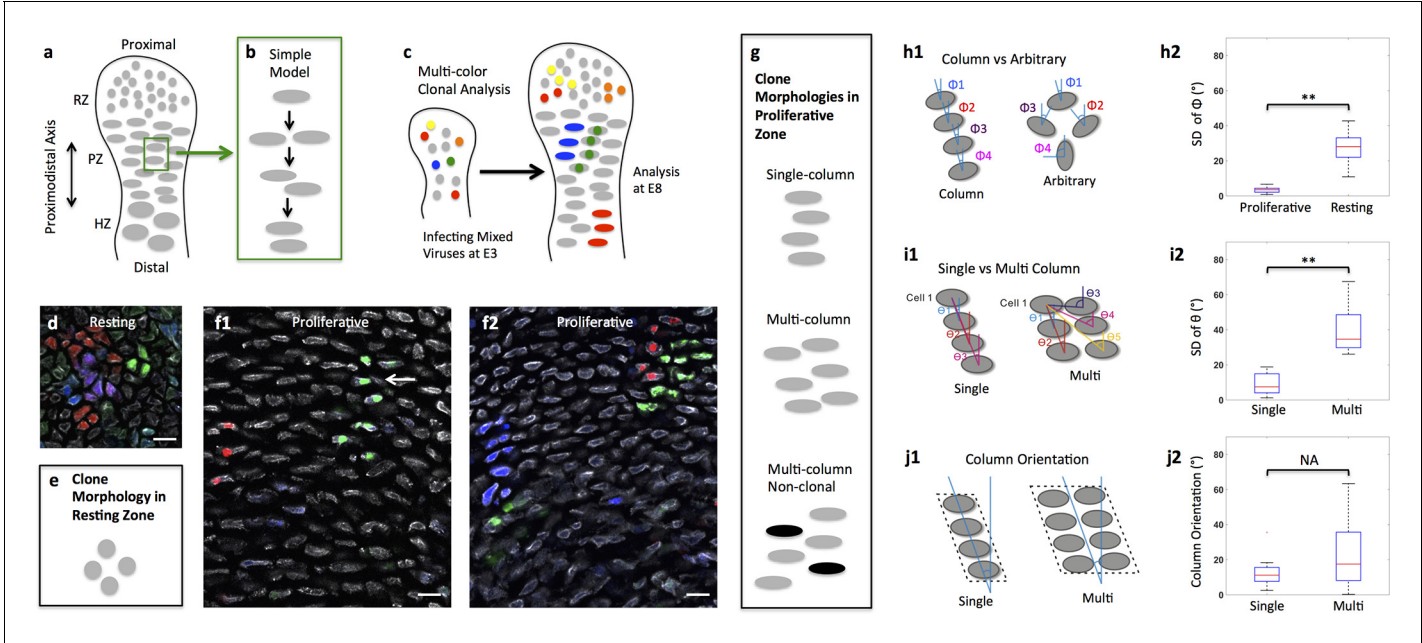

**Figure 1.** Diversity of clone morphology in growth plate cartilage. (a) Schematic diagram of growth plate cartilage. The tissue is comprised of three major growth zones along its proximodistal axis: resting (RZ), proliferative (PZ), and hypertrophic (HZ). The progenitor resting chondrocytes are spherical and dispersed whereas proliferative cells are ellipsoidal and more regularly arranged. The enlarged hypertrophic cells are terminally differentiated and are subsequently replaced by osteoblasts. (b) A simple model to explain column formation of proliferative cells involves oriented division orthogonal to the proximodistal axis followed by cell rearrangement. (c) The principle of viral-based multicolor clonal analysis. A mixture of recombinant replication-incompetent avian (RIA) retrovirus with distinct fluorescent markers is injected into chicken limb buds. If the simple model is correct, only single columns with distinct and uniform monocolor should be visualized in cartilage. (d–g) Distinct clone morphologies in the resting and the proliferative zones. Frozen tissue sections infected with RIA viruses expressing CFP (blue), GFP (green), membrane-GFP, mCherry (red) and H2B-mCherry (red) were counterstained with phalloidin Alexa-647 (gray), and imaged using confocal microscopy (d, f). Some clones displayed magenta or cyan due to viral coinfection of progenitor cells. While the growth direction of resting clones was arbitrary (d) (n = 30), proliferative clones were arranged in either single or multi-columns that appeared to be mainly in the growth direction (f1, f2) (n = 46). Some multi-columns were intermingled with neighboring clones (white arrow, f1). Schematic diagrams were drawn to highlight different clone morphologies (e, g). (h) Quantitative characterization of clone morphology. For individual clones with more than two cells, the angle (Φ) between the minor axis of each cell relative to the tissue proximodistal axis was measured (h1). The mean Φ in the proliferative and resting regions was 12° and 42°, respectively. See also *Figure 1—figure supplement 1*. The standard deviation (SD) of Φ in the proliferative zone was 3° (h2) (n = 18), smaller than cell orientation (12°), demonstrating these cells stacked in rows. See also *Figure 1—source data 1*. (i) Distinguishing single and multi-columns. Within individual proliferative columns, the orientation (θ) between the topmost cell (cell 1) and all the other cells was quantified (i1). Using 12° as a threshold, if the SD of θ was lower than this value, the column was grouped as a single column (n = 29); otherwise, we called it complex one (n = 20) (i2). See also *Figure 1—source data 2*. (j) Column orientation analysis. A polygon was drawn along the borders of columns, and the orientation of the major axis of the polygon to the proximodistal axis was measured (j1). While most single columns were generally parallel to the growth direction (mean orientation was 11.77°) (n = 21), multi-columns were slightly shifted with higher variations (mean orientation was 22.82°) (n = 21) (j2). See also *Figure 1—source data 3*. Scale bars: 15 µm. ** denotes p<0.01, NA denotes not significant (Wilcoxon Rank-Sum Test).

DOI: https://doi.org/10.7554/eLife.23279.002

The following source data and figure supplements are available for figure 1:

**Source data 1.** Quantitative characterization of clone morphology in wild-type tissues.
DOI: https://doi.org/10.7554/eLife.23279.005

**Source data 2.** Distinguishing single and multi-columns in wild-type tissues.
DOI: https://doi.org/10.7554/eLife.23279.006

**Source data 3.** Column orientation analysis in wild-type tissues.
DOI: https://doi.org/10.7554/eLife.23279.007

**Figure supplement 1.** Resting and proliferative chondrocytes exhibit distinct orientations.
DOI: https://doi.org/10.7554/eLife.23279.003

**Figure supplement 1—source data 1.** Characterizing chondrocyte cell orientation in wild-type tissues.
DOI: https://doi.org/10.7554/eLife.23279.004

(*Figure 1c*); alternatively, a polyclonal column would display multiple colors with salt-and-pepper distribution.

The results showed that limb cartilage was comprised of clustered cells with the majority displaying monocolor (*Figure 1d and f*), strongly suggesting that each cluster represents a clone. In the resting zone, the spherical and dispersed progenitor cells exhibited radial expansion (*Figure 1d, e and h*); in contrast, clonally related cells in the proliferative zone were arranged into either single columns one cell diameter in width or multi-column clones that were two or more cells wide (*Figure 1f, g, h and i*) (*Figure 1—figure supplement 1*). Multicolor cell tagging permitted us to further distinguish the boundaries between neighboring clones. Whereas all the single column clones (the red clone, *Figure 1f1*) and most multi-column clones (92%) (the blue and green clones, *Figure 1f2*) were monoclonal, a minority of multi-column clones (8%) intermixed with non-clonally related cells (white arrow pointing to the uninfected cell beside the green clone, *Figure 1f1*). Thus, proliferating cartilage is a mosaic of simple and multi-column monoclones, with a minor contribution resulting from intermixing between neighboring clones,

By measuring the orientation of columns in the proliferative zone, we further revealed that while single columns extended along the tissue proximodistal axis, multi-column clones were slightly shifted, but were still organized in contrast to the arbitrary arrangement of clones in the resting zone (*Figure 1j*). This raises an intriguing question regarding what mechanisms underlie polarized growth of proliferative clones. Below we address this question, focusing on regulation of single column formation.

## Single column formation is generated by cell pivot behavior

Proliferative cells divide orthogonal to the direction of growth (*Dodds, 1930*). Therefore, daughter cells must reorganize in order to stack into single columns oriented along the proximodistal axis. As a first step in exploring the cellular mechanisms driving this cell rearrangement, we followed live chondrocytes to determine their spatiotemporal dynamics. To clearly visualize dividing cells, we used a bicistronic RCASB (replication-competent avian retrovirus with B coat envelope protein) that simultaneously encodes H2B-GFP and mCherry to fluorescently label both nuclei and cytoplasm. Later, the infected metacarpals–small elements of limb cartilage—were mounted on homemade culture dishes (*Li et al., 2015*) for live imaging by one-photon confocal microscopy.

Using quantitative analyses of cell trajectories, we confirmed that chondrocytes undergo oriented divisions orthogonal to the proximodistal axis (*Figure 2a*) (*Figure 2—figure supplement 1*) (*Video 1*). In addition, we noted that after cytokinesis sister cells remained connected to each other (*Figure 2b and c*) and some started to reorient into columns (*Figure 2d*) (*Video 2*).

We observed many cell doublets whose tight association made it highly likely that they were sister cells. Focusing on these cell doublets, we noted that they were initially aligned with each other along the mediolateral axis of the tissue, consistent with the possibility that they had recently undergone cytokinesis (*Figure 2e*) (*Video 3*). Subsequently, one cell pivoted orthogonal to the other cell, resulting in stacking one atop the other (*Figure 2e, f and g*) (*Video 3*). We refer to this cell behavior as 'cell pivot' to reflect the morphological change over time (*Figure 2h*).

## Inhibiting PCP signaling disrupts oriented cell division, but not cell pivot behavior

Whereas cell intercalation has been proposed to underlie skeletal growth, very few studies have been performed on "cell pivot" behavior in the developing limb skeleton (*Ahrens et al., 2009*; *Le Pabic et al., 2014*; *Romereim et al., 2014*; *Li et al., 2015*) Therefore, we next probed how cell pivoting is regulated at molecular level. Previous work including our own work have demonstrated that cartilage and surrounding perichondrium express the main PCP components including Frizzled-7, Vangl-2, Dishevelled-2, and all are required for oriented division and cartilage tissue architecture (*Li and Dudley, 2009*; *Le Pabic et al., 2014*; *Gao et al., 2011*; *Sisson et al., 2015*; *Hartmann and Tabin, 2000*; *Sisson and Topczewski, 2009*; *Kuss et al., 2014*). However, all these studies have solely relied on static observations leaving open the questions of whether and how PCP signaling might influence cell rearrangement in space and time.

To analyze these questions in a dynamic manner, we turned to live imaging together with molecular perturbation. To this end, we took advantage of a truncated version of Frizzled-7 (Fzd7) lacking

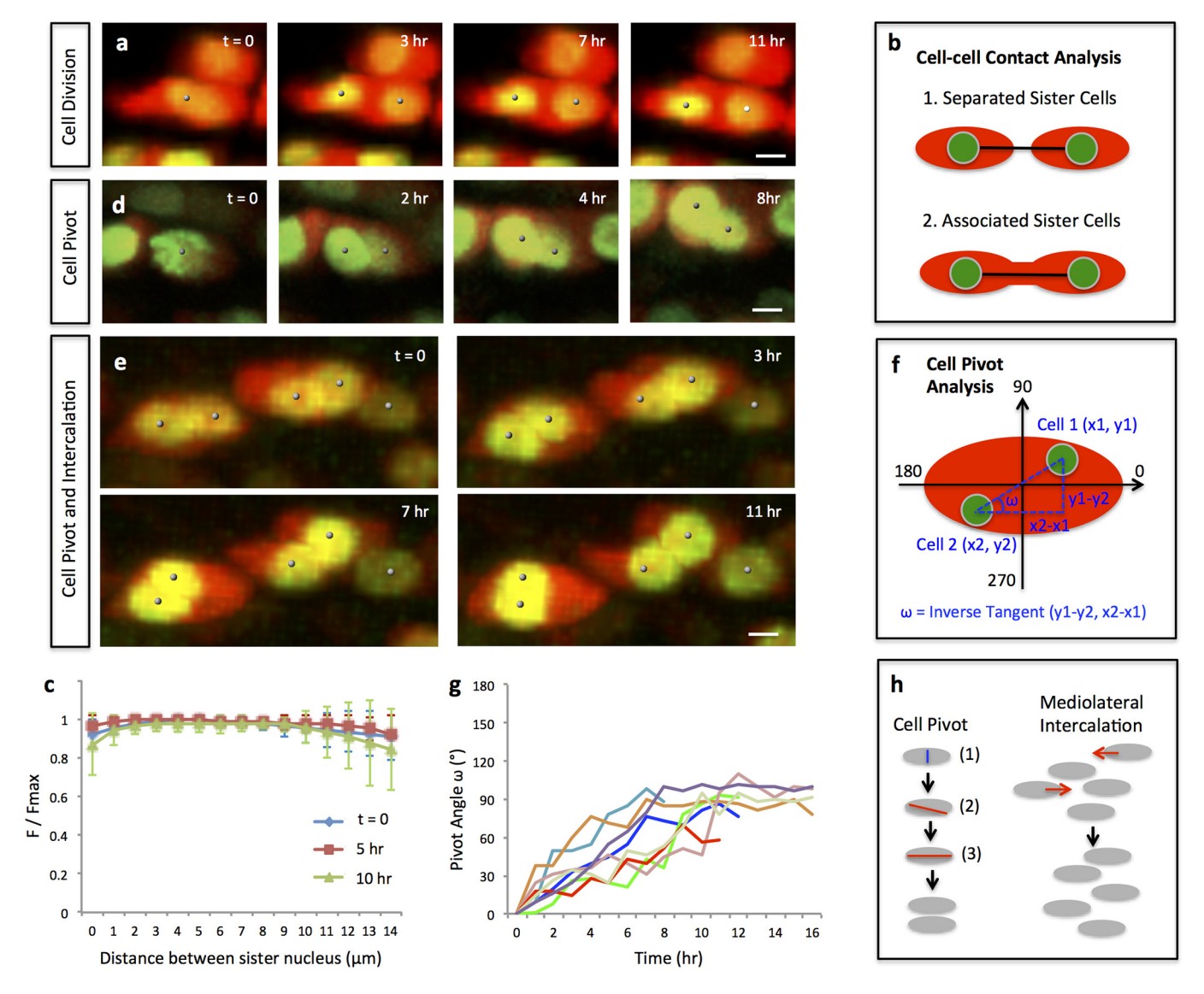

**Figure 2.** Single and multi-columns are generated by distinct cell rearrangements. (a) Oriented cell division in the proliferative zone. Live imaging was performed on the chick metacarpal explants expressing H2B-GFP (green) and mCherry (red) via replication-competent avian retroviral (RCAS) infection. Sister cells appeared to be positioned orthogonal to the tissue proximodistal axis and physically coupled after cytokinesis. See also *Figure 2—figure supplement 1*. (b, c) Quantitative analysis of sister cell contact. Polyline kymograph analysis of the cytoplasmic mCherry intensity (red) was conducted between the nucleus of two sister cells (green). The intensity was divided by the maximum intensity found alone the line for normalization (F/Fmax). For individual dividing pairs, T = 0 was the time when two daughter nuclei formed. If sister cells separate later (b-1), the F/Fmax curve should display an inversed bell shape; if the F/Fmax curve remains flat, it means the two cells are physically associated (b-2). This method confirmed sister cells remained in contact 10 hr after cytokinesis (n = 5) (c). See also *Figure 2—source data 1*. (d, e) Snapshots of cell pivot and mediolateral intercalation in the proliferative zone. In d, one mother cell produced two daughters started to rearrange their relative positions (The signal intensity of the images in d was adjusted from the corresponding movie to present nucleus morphologies more clearly). In e, on the left, two laterally aligned sister cells (cell doublets) reorganized their orientations and consequently stacked into a single column; on the right, three cells underwent intercalation: they were arranged laterally positioned at time 0; afterwards, the right and left intercalated toward the middle at 11 hr. (f) Schematic diagram to show the calculation of pivot angle (ω) that is between the plane of sister cells and the mediolateral axis of the tissue. Imaging software IMARIS was used to generate the coordinates of the cells (x, y) that were further applied to the provided equation to calculate ω at different time points. (g) Sister cells underwent pivot. ω of individual sister pairs was plotted against time and presented as individual lines in the graph. T = 0 was the time when sister cells were started to be observed, and ω at this time point was normalized to 0° for the ease of comparison. Most pairs (7/8) were initially aligned lateral to each other (low ω at time 0) and then underwent pivot (progressive increase of ω) into a single column (ω was between 70–100°) along the tissue proximodistal axis (n = 8). The maximal limit of y axis was set to 180° because the pivot angles of some cells were larger than 90°. Noticeably, one pair underwent partial pivot for about 50° only (red line). See also *Figure 2—source data 2*. (h) Schematic diagrams showing how distinct types of cell

*Figure 2 continued on next page*

*Figure 2 continued*

rearrangements generating corresponding columns: cell pivot behavior produces single columns whereas intercalation refines multi-columns. The blue and red lines stand for cleavage furrow and post-cleavage furrow, respectively. Scale bars: 4 μm.

DOI: https://doi.org/10.7554/eLife.23279.008

The following source data and figure supplements are available for figure 2:

**Source data 1.** Quantitative analysis of sister cell contact in wild-type tissues.

DOI: https://doi.org/10.7554/eLife.23279.014

**Source data 2.** Quantitative analysis of cell pivot in wild-type tissues.

DOI: https://doi.org/10.7554/eLife.23279.015

**Figure supplement 1.** Oriented cell division in the proliferative zone.

DOI: https://doi.org/10.7554/eLife.23279.009

**Figure supplement 1—source data 1.** Characterizing cell division orientation in wild-type tissues.

DOI: https://doi.org/10.7554/eLife.23279.016

**Figure supplement 2.** Mediolateral intercalation in the proliferative zone.

DOI: https://doi.org/10.7554/eLife.23279.010

the PDZ binding domain (Fzd7-ΔPDB) that functions as a dominant-negative mutant (*Kuss et al., 2014*) and disrupts normal chondrocyte cell polarity in developing chick cartilage (*Li and Dudley, 2009*). By co-infecting A coated virus (RCASA) harboring the Fzd7-ΔPDB mutant together with RCASB-H2B-GFP-2A-mCherry into chicken limbs, we achieved super-infection, such that many fluorescently labeled cells became rounder and disorganized (*Figure 3—figure supplement 1*) undergoing cell division in arbitrary directions rather than along the mediolateral axis (*Figure 3a*) (*Figure 3—figure supplement 2*) (*Video 4*). Interestingly, despite their misorientation, sister cells remained physically coupled (*Figure 3a and b*) and underwent pivot behavior (*Figure 3c and d*) (*Video 5*).

Given the above results, we predicted that Fzd7-ΔPDB expressing sister cells would still stack into rows but that these rows would not be aligned along the proximodistal axis. To test this prediction by clonal analysis in vivo, we infected chick cartilage with RIA viruses that tag Fzd7-ΔPDB positive clones with distinct fluorescent markers, and subsequently examined their morphologies in frozen tissue sections (*Figure 4a*). Focusing on clones with single cell diameter, they were stacked (*Figure 4b and c*), but rather than forming columns in the direction of tissue growth, clone orientation was largely arbitrary (*Figure 4d*), suggesting that PCP pathway is required for normal oriented division but not for cell pivoting.

As a second method of perturbing the PCP pathway, we used a dominant-negative mutant of Dishevelled-2 that lacks the PDZ domain (DVL2-ΔPDB) such that it strongly inhibits the PCP pathway with little or no effect on canonical Wnt signaling (*Li and Dudley, 2009*). The cell phenotypes observed after DVL2-ΔPDB treatment were similar to those in Fzd7-ΔPDB expressing tissues (*Figure 3e–h*) (*Video 6*) (*Video 7*). Further supporting a specific roles of PCP pathway in controlling this polarized cell behavior of chondrocytes, previous perturbation analyses of β-catenin in both chick and mouse cartilage ruled out a function for canonical Wnt pathway in oriented cell division or cell morphology (*Li and Dudley, 2009*; *Ahrens et al., 2011*).

## Activating PCP signaling disrupts both oriented cell division and cell pivot behavior

One unique feature of PCP signaling in vertebrates is that gain- and loss-of-

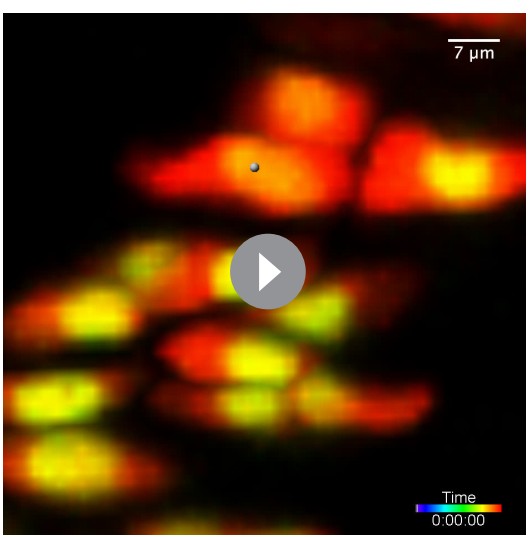

**Video 1.** Live imaging of oriented cell division in the chick metacarpal expressing H2B-GFP (green) and mCherry (red). The snapshots of the segmented cells (white dots) were presented in *Figure 2a*. Scale bar: 7 μm

DOI: https://doi.org/10.7554/eLife.23279.011

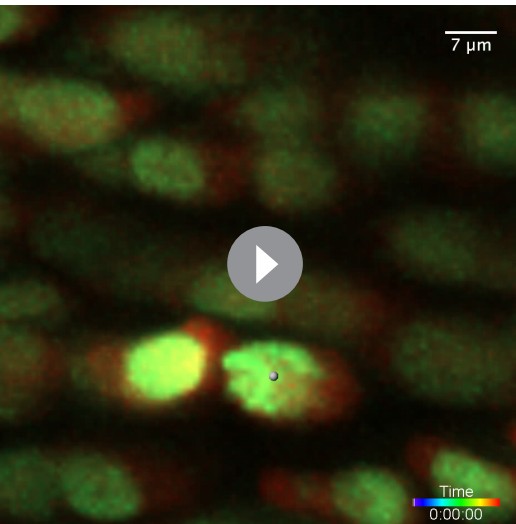

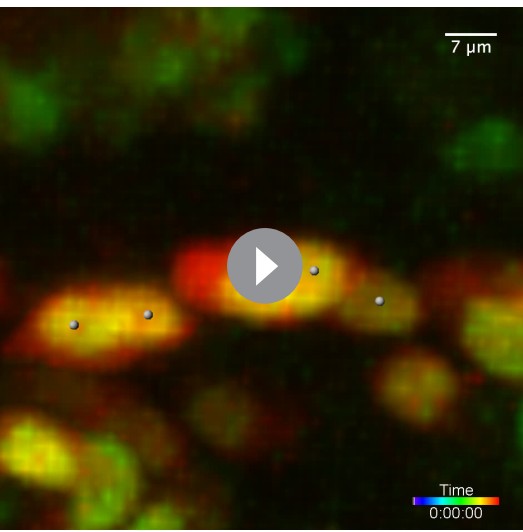

**Video 2.** Live imaging on the chick metacarpal expressing H2B-GFP (green) and mCherry (red). One mother cell generated two daughter cells that underwent partial pivot. The snapshots of the segmented cells (white dots) were presented in *Figure 2d*. Scale bar: 7 μm
DOI: https://doi.org/10.7554/eLife.23279.012

**Video 3.** Live imaging of two pivoting cells (left) and three intercalating cells (right) in the chick metacarpal expressing H2B-GFP (green) and mCherry (red). The snapshots of the segmented cells (white dots) were presented in *Figure 2e*. Scale bar: 7 μm
DOI: https://doi.org/10.7554/eLife.23279.013

function analyses produce similar phenotypes (*Park et al., 2005*). One interpretation of this observation is that the levels of PCP proteins on the cell membrane must be tightly regulated for proper signal transduction (*Park et al., 2005*). To test this, we asked whether activating PCP signaling would phenocopy the loss-of-function phenotype in cartilage. To this end, we utilized either full-length Fzd7 or Vangl2, which is membrane-bound in cartilage (*Gao et al., 2011*), to promote PCP activity. RCAS expression of either transgene disrupted normal cell morphology (*Figure 3—figure supplement 1*) and oriented division in developing cartilage (*Figure 3i and k*) (*Figure 3—figure supplement 2*) (*Video 8*) (*Video 9*), similar to loss-of-function phenotype; however, sister cells became separated and failed to stack into columns (*Figure 3j and l*). Accordingly, clonally related cells expressing Fzd7 were arbitrarily arranged (*Figure 4e and f*).

Together with the loss-of-function studies, these results show that oriented division is sensitive to both high and low PCP activity, whereas cell pivot behavior is only inhibited by high PCP activity (*Figure 4g*). Hence, these two cellular events are coupled but respond differently to the level of PCP signaling.

### Excess membrane-bound Frizzled-7 receptor inhibits cell pivot behavior

The finding that gain (Fzd7) versus loss (Fzd7-ΔPDB) of PCP signaling cause opposite effects on cell pivot behavior raised the intriguing possibility that the PDB domain may play a role in this process. In epithelia cells, this domain is essential for the subcellular localization and signaling specificity of Fzd (*Wu et al., 2004*). To test this possibility in chondrocytes, we introduced either Fzd7-YFP or Fzd7-ΔPDB-YFP fusion into chick limbs, using the YFP marker since good antibodies to chick Fzd7 were not available. Interestingly, Fzd7-YFP was distributed on both the cell membrane and within the cytoplasm (*Figure 5a and f*) (*Figure 5—figure supplement 1*) whereas Fzd7-ΔPDB-YFP was largely localized to the cytoplasm (*Figure 5b*. 5 f). These results suggest that cell pivot behavior is blocked by excess membrane-bound PCP components.

Because membrane localization of Fzd7 is determined by its PDB domain that contains two motifs (KTxxxW and VTTE) to mediate protein-protein interaction (*Wong et al., 2003*) (*Figure 5e*), we next sought to tease apart the roles of these two motifs by removing each of them individually from Fzd7 and examining the subsequent effect on Fzd7 localization. Whereas the KTxxxW truncated version displayed membrane and cytoplasmic distribution similar to that of intact Fzd7 (data not shown), the

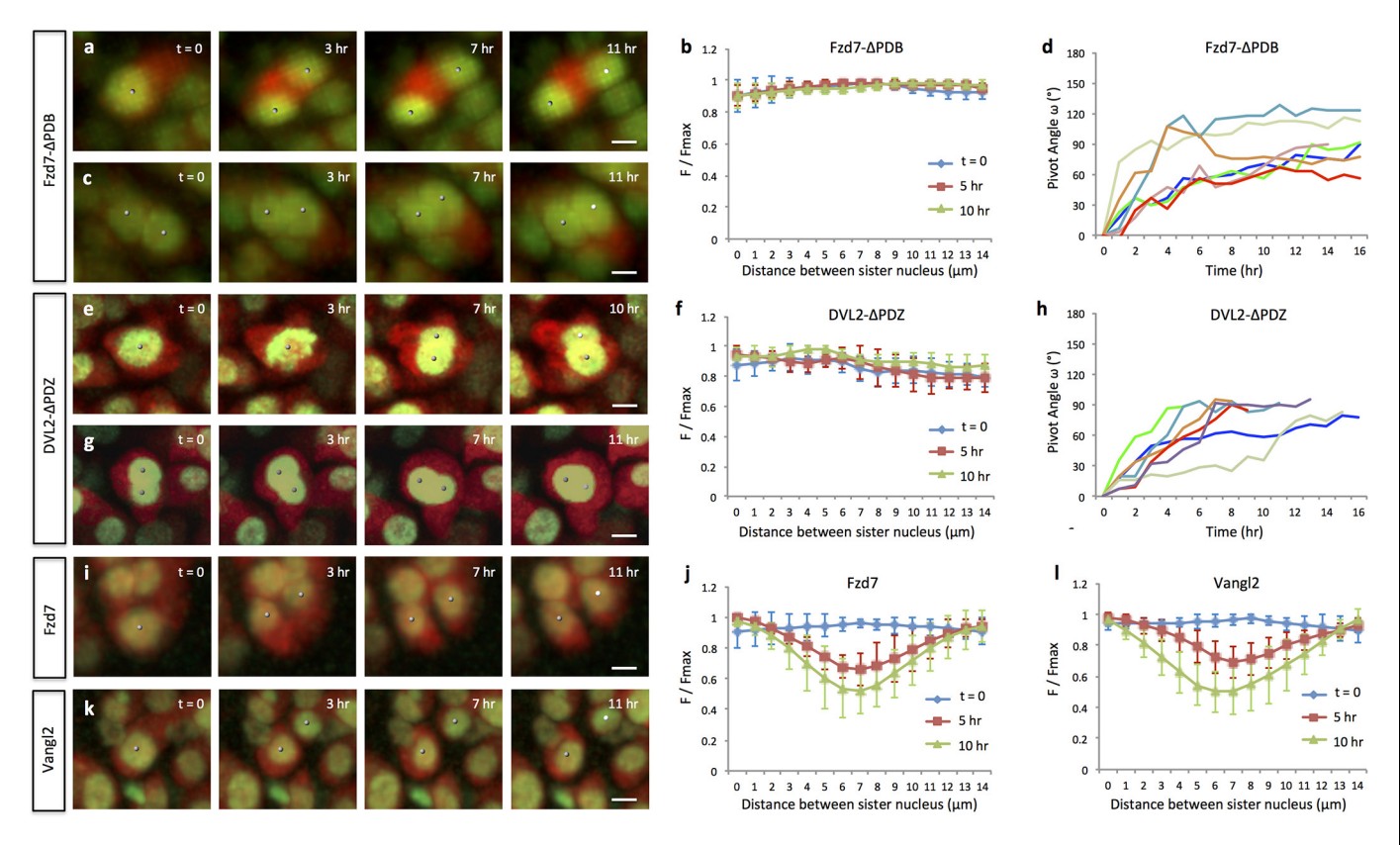

**Figure 3.** Oriented cell division and cell pivot are differentially regulated by PCP signaling. (**a, b**) Misoriented division and physical coupling of sister cells in the presence of Fzd7-ΔPDB. Live imaging was performed on the chick metacarpals expressing H2B-GFP (green), mCherry (red) and Fzd7-ΔPDB via RCAS infection (**a**). mCherry intensity analysis confirmed the sister cells were connected after cytokinesis (n = 5) (**b**). See also *Figure 3—source data 1*. (**c, d**) Normal pivot behavior of Fzd7-ΔPDB expressing cells. Among 18 cell doublets, seven pairs underwent pivot to form single stacks. Though their division orientation was not along the mediolateral axis (See also *Figure 3—figure supplement 2*), these orientation (T = 0) was normalized to 0° for the ease of comparison with wild-type cells in *Figure 2g*. Note one pair underwent partial rearrangement after division (red line). See also *Figure 3—source data 2*. (**e–h**). In the tissues with exogenous DVL2- ΔPDZ, sister cells were associated after cytokinesis (**e, f**) (n = 5) and 35% cell doublets were rearranged (**g, h**) (n = 20). The signal intensity of the images in g was adjusted from the corresponding movie to present nuclei morphologies more clearly. See also *Figure 3—source data 3*, *Figure 3—source data 4*. (**i–l**) In the tissues expressing Fzd7 (**i, j**) or Vangl2 (**k, l**), cells did not divide along the mediolateral axis (**i, k**), and were separated within 5 hr after cytokinesis (n = 5 for both cases) (**j, l**). No rearrangement was observed between cell doublets in the presence of Fzd7 (n = 18) or Vangl2 (n = 16). See also *Figure 3—source data 5*, *Figure 3—source data 6*. Scale bars: 4 μm.

DOI: https://doi.org/10.7554/eLife.23279.017

The following source data and figure supplements are available for figure 3:

**Source data 1.** Quantitative analysis of sister cell contact in Fzd7-ΔPDB expressing tissues.
DOI: https://doi.org/10.7554/eLife.23279.022

**Source data 2.** Quantitative analysis of cell pivot in Fzd7-ΔPDB expressing tissues.
DOI: https://doi.org/10.7554/eLife.23279.023

**Source data 3.** Quantitative analysis of sister cell contact in DVL2-ΔPDZ expressing tissues.
DOI: https://doi.org/10.7554/eLife.23279.024

**Source data 4.** Quantitative analysis of cell pivot in DVL2-ΔPDZ expressing tissues.
DOI: https://doi.org/10.7554/eLife.23279.025

**Source data 5.** Quantitative analysis of sister cell contact in Fzd7 expressing tissues.
DOI: https://doi.org/10.7554/eLife.23279.026

**Source data 6.** Quantitative analysis of sister cell contact in Vangl2 expressing tissues.
DOI: https://doi.org/10.7554/eLife.23279.027

**Figure supplement 1.** Chondrocytes with perturbed PCP activity display abnormal morphologies.
DOI: https://doi.org/10.7554/eLife.23279.018

**Figure supplement 1—source data 1.** Characterizing cell orientation in wild-type, Fzd7, Fzd7-ΔPDB or Vangl2 expressing tissues.

*Figure 3 continued on next page*

*Figure 3 continued*

DOI: https://doi.org/10.7554/eLife.23279.020

**Figure supplement 2.** Chondrocytes with perturbed PCP activity display misoriented division.

DOI: https://doi.org/10.7554/eLife.23279.019

**Figure supplement 2—source data 1.** Characterizing cell division orientation in Fzd7-ΔPDB, Fzd7 or Vangl2 expressing tissues.

DOI: https://doi.org/10.7554/eLife.23279.021

VTTE truncated mutant was constrained to the cytoplasm (*Figure 5c and f*), similar to Fzd7-ΔPDB-YFP. Hence, the VTTE motif is critical for membrane localization of Fzd7-YFP. Further supporting this, we found that adding the VTTE motif onto the C-terminus of Fzd7 promoted membrane binding (*Figure 5d and f*) (*Figure 5—figure supplement 1*).

## N-Cadherin is enriched in the post-cleavage furrow

Given that high expression levels of Fzd7 result in sister cell separation, we asked whether there might be disruption of cell adhesion molecules. PCP proteins have been shown to promote internalization of cadherins from the membrane and adherens junctions in epithelial cells (*Warrington et al., 2013*; *Nagaoka et al., 2014*). Additionally, cadherins are important to normal limb chondrogenesis and craniofacial skeletal growth (*Oberlender and Tuan, 1994*; *Romereim et al., 2014*). Hence, we hypothesized that in growth plate cartilage, excess membrane-bound Fzd might prevent cell pivoting by reducing local cadherin concentration.

To test this hypothesis, we first examined the expression pattern of N-cadherin (Ncad) in the chick limb cartilage. Immunofluorescence demonstrated that Ncad was mainly distributed on the membrane with small amounts within the cytoplasm of interphase cells; however, it was greatly enriched between closely associated cells (*Figure 6a*, *Romereim et al., 2014*). Interestingly, counter-staining of contractile rings demonstrated that this sub-population of Ncad was not located in the cleavage furrow (*Figure 6—figure supplement 1*), but rather concentrated between closely associated cells that appeared to be at post-cleavage stage.

We next examined the spatiotemporal dynamics of Ncad expression. To this end, the limb was infected with low-titer RCAS expressing Ncad-GFP fusion. After infection, the developing limbs displayed typical tissue architecture and morphology (data not presented), suggesting low levels of exogenous Ncad did not significantly impact normal growth. Live imaging revealed a distribution pattern of Ncad-GFP similar to that of endogenous Ncad, with expression on the membrane and inside the cytoplasm (*Figure 6e*) (*Video 10*). Importantly, fluorescence intensity of junctional Ncad-GFP was enriched between sister cells and subsequently maintained at a similar level during cell pivot into single columns (*Figure 6e and h*). These findings demonstrate that Ncad is concentrated at the cell membrane in the post-cleavage furrow.

## N-cadherin is required for sister cell association and cell pivot behavior

The tight correlation between junctional Ncad and cell behavior suggests that formation of adherens junctions between sister cells after cytokinesis may be critical for subsequent cell rearrangement. To test this possibility, we imaged metacarpal cultures treated with an

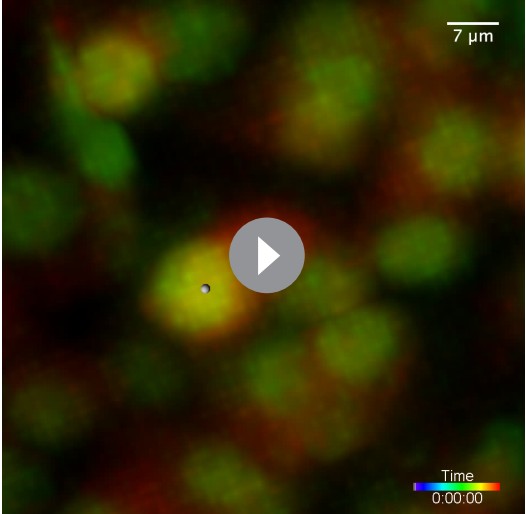

**Video 4.** Live imaging of misoriented cell division in the chick metacarpal expressing H2B-GFP (green), mCherry (red) and Fzd7-ΔPDB. After division, the sister cells remained connected to each other. The snapshots of the segmented cells (white dots) were presented in *Figure 3a*. Scale bar: 7 μm

DOI: https://doi.org/10.7554/eLife.23279.028

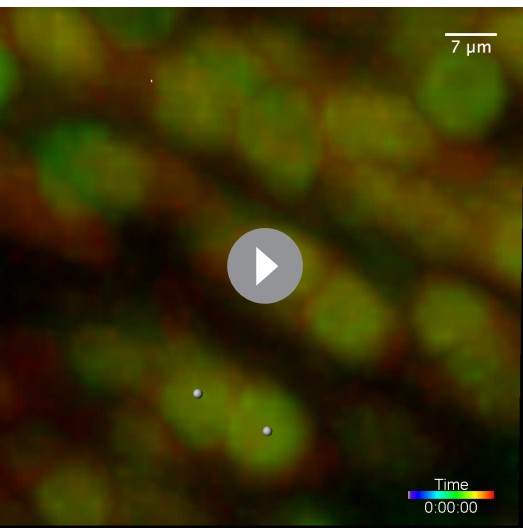

**Video 5.** Live imaging of two pivoting cells in the chick metacarpal expressing H2B-GFP (green), mCherry (red) and Fzd7-ΔPDB. The snapshots of the segmented cells (white dots) were presented in *Figure 3c*. Scale bar: 7 μm

DOI: https://doi.org/10.7554/eLife.23279.029

antibody that has previously been shown to effectively block N-cadherin function in limb micromass cultures (*Oberlender and Tuan, 1994*; *Gänzler-Odenthal and Redies, 1998*). We titrated the supernatant containing this antibody and found that chondrocytes in cartilage explants incubated in medium supplemented with a 1:10 dilution of the supernatant were disorganized (*Video 11*); under this culture condition, after cytokinesis daughter cells subsequently separated and failed to stack into columns (*Figure 6i and j*).

In order to confirm the role of Ncad, we ectopically expressed a truncated dominant-negative Ncad mutant (dnNcad) that lacks the extracellular domain (*Kintner, 1992*), and observed similar cellular phenotypes (*Figure 6k and l*) (*Video 12*). These data demonstrate that Ncad is functionally involved in chondrocyte cell association and rearrangement (*Figure 6m*).

## Influence of PCP signaling on cell pivot behavior involves junctional N-Cadherin

The defective cell rearrangement after blocking Ncad function is similar to that observed after enhancing PCP activity, prompting us to further test their causal linkage. In the tissues expressing either Fzd7 or Fzd7-ΔPDB, Ncad transcript levels appeared normal (*Figure 6—figure supplement 2*). Furthermore, immunofluorescence analysis failed to detect obvious differences in the amount of membrane and cytoplasmic-bound Ncad protein when compared with wild-type tissues (*Figure 6a–c*). In contrast, junctional Ncad was diminished in the Fzd7 expressing tissues (*Figure 6c and d*), while normal in Fzd7-ΔPDB expressing tissues (*Figure 6b and d*), consistent with the phenotypes observed after PCP and Ncad perturbation studies.

To address how PCP affects the dynamics of junctional Ncad, we performed intensity analysis of junctional Ncad-GFP in developing cartilage. In the presence of Fzd7, Ncad was concentrated in the post-cleavage furrows of the misoriented dividing cells and subsequently maintained at the interface between rearranging sister cells (*Figure 6f and h*) (*Video 13*), similar to wild-type cells (*Figure 6e and h*) (*Video 10*). However, in the cells with high intact Fzd7, Ncad was initially enriched in the post-cleavage furrows but its local concentration became reduced over time (*Figure 6g and h*) (*Video 14*).

The fact that Ncad-GFP labels the cell membrane (*Figure 6e–g*) provides a clean means for performing cell segmentation and quantitation to validate sister-cell association. Therefore, we further measured fluorescence intensity changes of this fusion protein across both sisters cell during their rearrangement (*Figure 6—figure supplement 3a*). The results show that Ncad-GFP was enriched in the junction shortly after cell division. At subsequent times, Ncad was still maintained at cell junctions of wild-type and Fzd7-ΔPDB expressing cartilage (*Figure 6—figure supplement 3b–d*), but vanished from Fzd7 expressing cartilage. Such patterns were consistent with fluorescence intensity changes of cytoplasmic mCherry during cell pivoting (*Figures 2c*, *3b and j*), confirming physical connection of sister cells in wild-type and Fzd7-ΔPDB expressing tissues, but not after Fzd7 expression. Collectively, our results suggest that PCP signaling inhibits sister cell association and their pivoting via down-regulation of junctional cadherin.

In summary, we demonstrate that sister chondrocyte cells, regardless of their orientation during cell division, fail to reorganize into single columns if they do not attach to each other. Although both oriented division and cell rearrangement occur sequentially and are controlled by PCP signaling, the division orientation is not sufficient to initiate cell pivot behavior; rather, cell association mediated by Ncad is critical for their subsequent rearrangement. Taken together, we propose that the

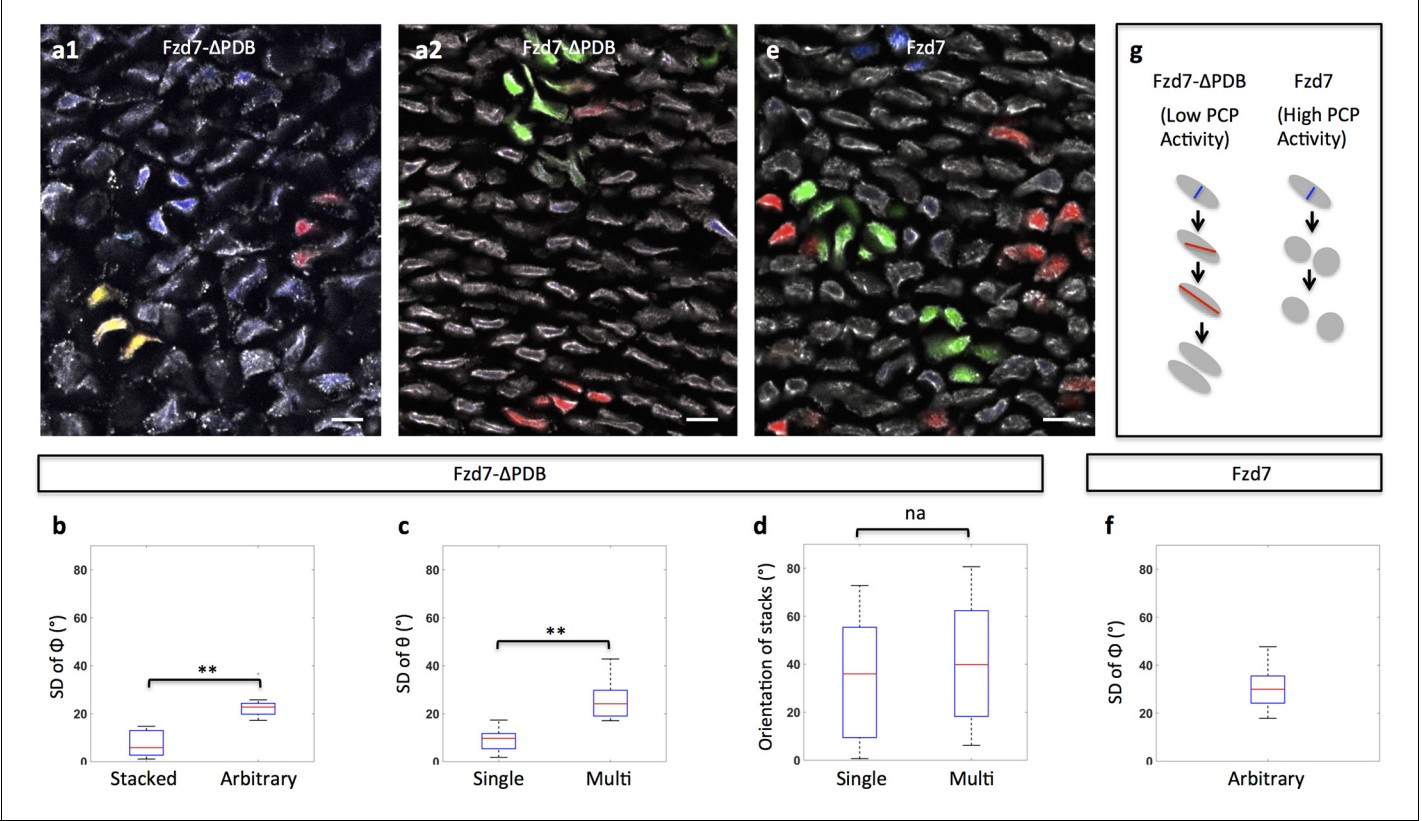

**Figure 4.** Clonal analysis confirms the roles of PCP signaling. (a) Recombinant viruses RIA-Fzd7-ΔPDB-T2A-CFP, RIA-Fzd7-ΔPDB-T2A-mCherry and RIA-Fzd7-ΔPDB-T2A-YFP were mixed and injected into chick limb buds; at later stages, clone morphologies were examined in the frozen cartilage sections. An arbitrarily arranged clone (blue, **a1**), single-row clone with normal (red, **a1**) or shifted (yellow, **a1**) orientation, multi-row clones (**a2**) were observed. (b–d) Quantitative analysis of Fzd7-ΔPDB positive clone morphology. The same method in *Figure 1h–j* was applied. Briefly, the standard deviation (SD) of the angles between clonally related cells to the tissue proximodistal axis (Φ) was calculated to identify stacked (n = 22) (SD <12°) and arbitrarily arranged clones (n = 14) (SD >12°) (**b**). Within each stacked clone, we further measured the SD of θ, the orientation of the topmost cell relative to all the other cells. The results showed that these clones exhibited either single (n = 22) or complex (n = 12) width (**c**) and they were not parallel to the proximodistal axis (**d**) (n = 13 and 8 for single and multiple stacks, respectively). See also *Figure 4—source data 1*, *Figure 4—source data 2*, *Figure 4—source data 3*. (e, f) In the tissues expressing Fzd7-T2A-CFP, Fzd7-T2A-mCherry and Fzd7-T2A-YFP at clonal density, the clonally related cells appeared to be arbitrarily oriented without forming stacks (**e**). Consistently, the SD of the angles between these cells to the proximodistal axis (Φ) was larger than 12° (n = 16) (**f**). See also *Figure 4—source data 4*. (g) Schematic diagrams to illustrate different cell pivot behaviors under various perturbation conditions. With low PCP activity, cells divide in a misoriented manner followed by sister cell association and pivot. In contrast, with enhanced PCP activity, while cells still undergo misoriented divisions, the sister cells become separated and fail to rearrange. Scale bars: 15 μm. ** denotes p<0.01, NA denotes not significant (Wilcoxon Rank-Sum Test).

DOI: https://doi.org/10.7554/eLife.23279.030

The following source data is available for figure 4:

**Source data 1.** Quantitative characterization of clone morphology in Fzd7-ΔPDB expressing tissues.
DOI: https://doi.org/10.7554/eLife.23279.031
**Source data 2.** Distinguishing single and multiple stacks in in Fzd7-ΔPDB expressing tissues.
DOI: https://doi.org/10.7554/eLife.23279.032
**Source data 3.** Stack orientation analysis in Fzd7-ΔPDB expressing tissues.
DOI: https://doi.org/10.7554/eLife.23279.033
**Source data 4.** Quantitative characterization of clone morphology in Fzd7 expressing tissues.
DOI: https://doi.org/10.7554/eLife.23279.034

occurrence of cell pivot is enabled by cell-cell adhesion downstream of PCP signaling whereas the orientation of the cell pivot is determined by the orientation of preceding division (*Figure 6n*).

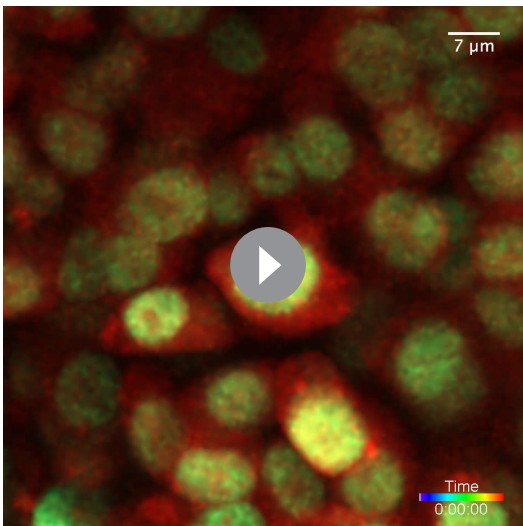

**Video 6.** Live imaging of misoriented cell division in the chick metacarpal expressing H2B-GFP (green), mCherry (red) and DVL2-ΔPDZ. After division, the sister cells were still associated. The snapshots of the segmented cells (white dots) were presented in *Figure 3e*. Scale bar: 7 μm

DOI: https://doi.org/10.7554/eLife.23279.035

## Multi-column clone formation appears to involve mediolateral cell intercalation

In addition to single column clones on which we focused above, we also observed patches of more than two cells in width in the sparsely RCAS infected tissues. Live imaging suggests that some cartilage cells move mediolaterally, in the process of intercalating between neighboring cells (*Figure 2e*) (*Video 3*). As a consequence, patches became elongated along the proximodistal direction, similar to a well-recognized mode of cell rearrangement that occurs during zebrafish jaw growth (*Le Pabic et al., 2014*).

To further test if the intercalating cells were clonally related and define their spatial-temporal relationships, we examined tissues infected by multicolored RIA viruses. Previous dynamic imaging using two-photon microscopy showed that chondrocyte cell intercalation is a slow process taking more than 48 hr to reach completion (*Li et al., 2015*). Due to technical constraints involving multiple fluorophore excitation, here we employed one-photon laser that can noninvasively image cartilage explant for up to 24 hr. Although this time period is not sufficient to follow complete intercalation, we observed partial intercalation within the same multi-column clone (*Figure 2—figure supplement 2a*) (*Video 15*) (80%), as well as some examples of intercalation between non-clonally related cells (*Figure 2—figure supplement 2b*) (*Video 16*) (20%). This existence of both cell intercalation and cell pivoting within clones (*Figure 2h*) suggests that

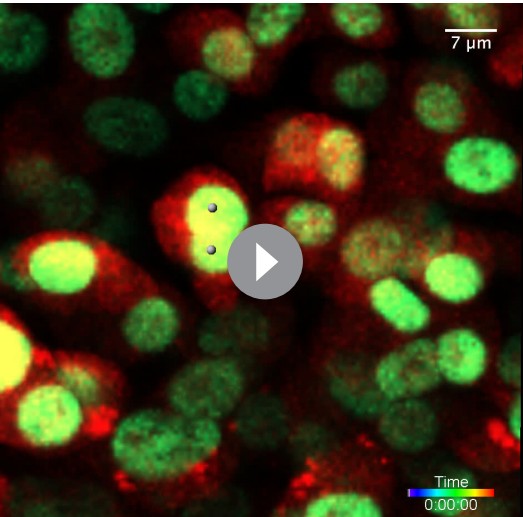

**Video 7.** Live imaging of two pivoting cells in the chick metacarpal expressing H2B-GFP (green), mCherry (red) and DVL2-ΔPDZ. The snapshots of the segmented cells (white dots) were presented in *Figure 3g*. Scale bar: 7 μm

DOI: https://doi.org/10.7554/eLife.23279.036

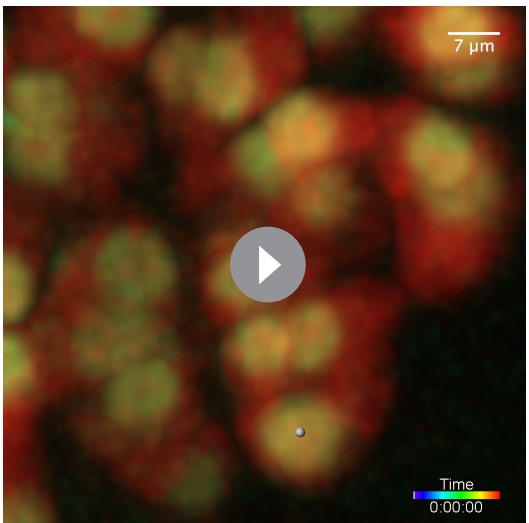

**Video 8.** Live imaging of misoriented cell division and sister cell separation in the chick metacarpal expressing H2B-GFP (green), mCherry (red) and Fzd7. The snapshots of the segmented cells (white dots) were presented in *Figure 3i*. Scale bar: 7 μm

DOI: https://doi.org/10.7554/eLife.23279.037

chondrocytes employ multiple strategies to establish proper tissue architecture.

## Discussion

Previous work on craniofacial cartilage in zebrafish and mouse has provided different interpretations regarding the cellular mechanisms driving chondrocyte cell stacking (*Sisson et al., 2015*; *Romereim et al., 2014*). It is notable that the zebrafish work was performed on Meckel's cartilage (*Sisson et al., 2015*) whereas the mouse study was focused on presphenoid synchondrosis (*Gänzler-Odenthal and Redies, 1998*), raising the possibility that different means of cell rearrangements occur in different types of skeletal elements and/or at different developmental stages. By combining live imaging and quantitative analyses in chick embryo, we demonstrate that cell pivoting contributes to normal morphogenesis of growth plate cartilage and further confirm the occurrence of mediolateral intercalation within the same context.

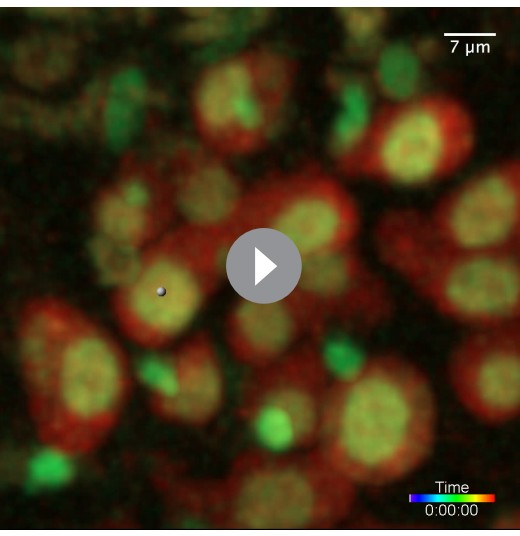

**Video 9.** Live imaging of misoriented cell division and sister cell separation in the chick metacarpal expressing H2B-GFP (green), mCherry (red) and Vangl2. The snapshots of the segmented cells (white dots) were presented in *Figure 3k*. Scale bar: 7 μm
DOI: https://doi.org/10.7554/eLife.23279.038

Our findings offer new insights into the relationship between oriented cell division and cell rearrangement. The association between these two cellular behaviors has been observed during mouse ureteric bud development and chick gastrulation (*Packard et al., 2013*; *Firmino et al., 2016*). In both cases, after cytokinesis, one daughter cell remains in its original position while the other disperses and reinserts at a position several cell diameters away, thus contributing to tissue elongation (*Packard et al., 2013*; *Firmino et al., 2016*). In contrast, chondrocytes reorganize within spatially constrained clones by pivot or intercalation behavior into oriented and organized columns in the direction of limb growth. The similarities in chondrocyte behavior in the developing craniofacial (neural crest-derived) and limb (mesoderm-derived) skeletal elements suggest that, during evolution, conserved cellular principles are employed to build cylindrical-shaped skeletons, even though these tissues are derived from different types of progenitor cells.

By delving into the molecular mechanisms underlying cell pivot behavior, we demonstrate that PCP signaling differentially influences distinct steps, regulating the orientation of cell division and enabling cells to pivot by controlling Ncad enrichment in the post-cleavage furrow. By this means, chondrocytes divide along the tissue mediolateral axis (cell major axis) due to geometrical constraints (*Li and Dudley, 2009*) while the resultant clones expand mainly in the direction of tissue growth. Given that proliferative chondrocytes normally are embedded in Collagen 2a and that cells become disorganized in its mutant mice (*Barbieri et al., 2003*), an intriguing possibility is that the extracellular matrix provides such a constraint.

By devising multicolor replication incompetent retroviruses for clonal analysis of chondrocytes, our work enriches the toolkit for cell lineage analysis in amniotes. Previous studies in the chick embryo have utilized sparse labeling with low-titer RIA viruses harboring a single histological marker (*Chen et al., 1999*). While this method has the advantage of permanently labeling cells and their progeny, it has limitations. First, in order to unambiguously infect a single progenitor, the virus needs to be highly diluted such that individual labeled clones are distant from each other; however, this low labeling density prevents assessment of interactions between neighboring clones. Second, visualizing markers within clones requires histological processing of the specimen that can affect cellular morphology. To meet these two challenges, we have created an RIA based fluorescent labeling technique to achieve combinatorial tagging of multiplex clones, permitting determination of clonal relationship in intact tissues with relatively high labeling density. Compared with Confetti techniques mediated by lentiviral delivery (*Loulier et al., 2014*), the RIA virus offers better levels of infection in

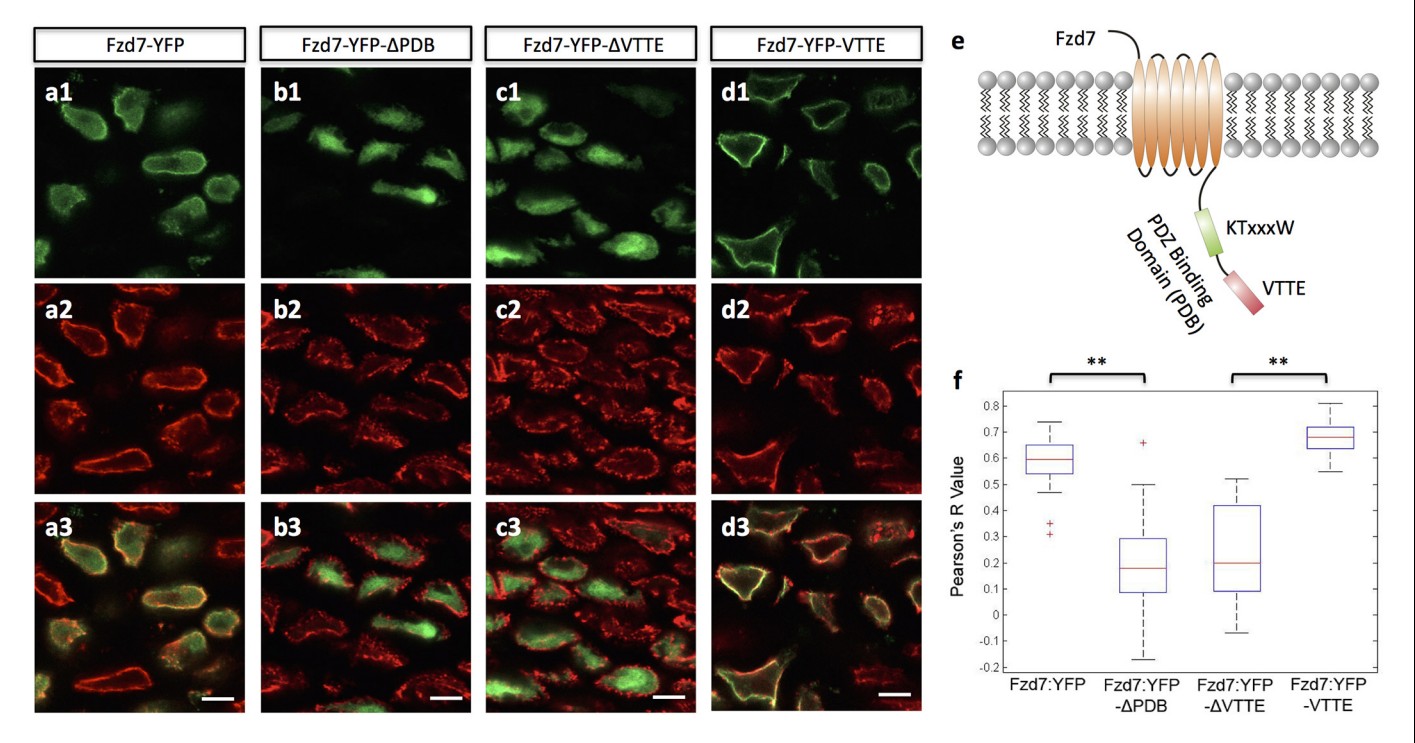

**Figure 5.** The subcellular localization of Frizzled-7 is determined by its PDB domain. (a–d) VTTE motif is essential to the membrane localization of Fzd7. Chick humerus expressing different versions of Fzd7-YFP (green) was sectioned and counterstained with phalloidin Alexa-647 (red). Fzd7 was expressed in the cytoplasm and on the membrane (a) (n = 20), but was constrained to the cytoplasm when lacking its PDB domain (b) (n = 29). Removing the VTTE motif from and adding it onto Fzd7 promoted protein cytoplasmic and membrane localization, respectively (c, d) (n = 15 and 33 in d and e, respectively). See also *Figure 5—figure supplement 1*. (e) Schematic diagram of the PDB domain of Fzd7: two motifs KTxxxW and VTTE are predicted to bind to other signaling proteins. (f) Subcellular localization of different Fzd7 mutants were examined by calculating Pearson correlation coefficient to quantify the colocalization between YFP and phalloidin (one represents perfect correlation and −1 represents perfect anti-correlation). See also *Figure 5—source data 1*. Scale bars: 5 μm. ** denotes p<0.01(Wilcoxon Rank-Sum Test).

DOI: https://doi.org/10.7554/eLife.23279.039

The following source data and figure supplement are available for figure 5:

**Source data 1.** Pearson correlation analysis of Fzd7 mutants and phalloidin colocalization.
DOI: https://doi.org/10.7554/eLife.23279.041

**Figure supplement 1.** Ectopic expressed Fzd7 and Fzd7-VTTE are bound to cell membrane.
DOI: https://doi.org/10.7554/eLife.23279.040

many chick tissues such as retina and limb (*Pearse et al., 2007*), muscle (*Gordon et al., 2009*), skin and feather (*Li et al., 2013*). Importantly, as a VSV-G pseudotyped virus that infect all types of cells, our viral reagents can be easily adapted to study other morphogenetic events in both genetic and non-genetic animal models.

In this study, we dissect the complex cellular behavior driving limb skeletal elongation. Our results show that clonally related chondrocytes become arranged in either single or multi-columns along the axis of tissue elongation. These columnar morphologies are generated through two types of cell behaviors: cell pivoting resulting in single columns or cell intercalation resulting in complex columns. The cell pivoting exhibits certain similarities to cell rearrangement in mouse presphenoidal syn-chondrosis, a type of neural crest-derived craniofacial cartilage (*Romereim et al., 2014*). The PCP pathway coordinates cell pivoting following mediolateral division. In this way, chondrocytes increase their cell numbers while concomitantly undergoing stereotypical arrangements that result in tissue elongation. Together with the known roles of PCP signaling in determining the morphology of other tissues and organs (*Zallen, 2007*), this highlights the importance of PCP signaling in shaping tissues via regulating polarized cell behaviors.

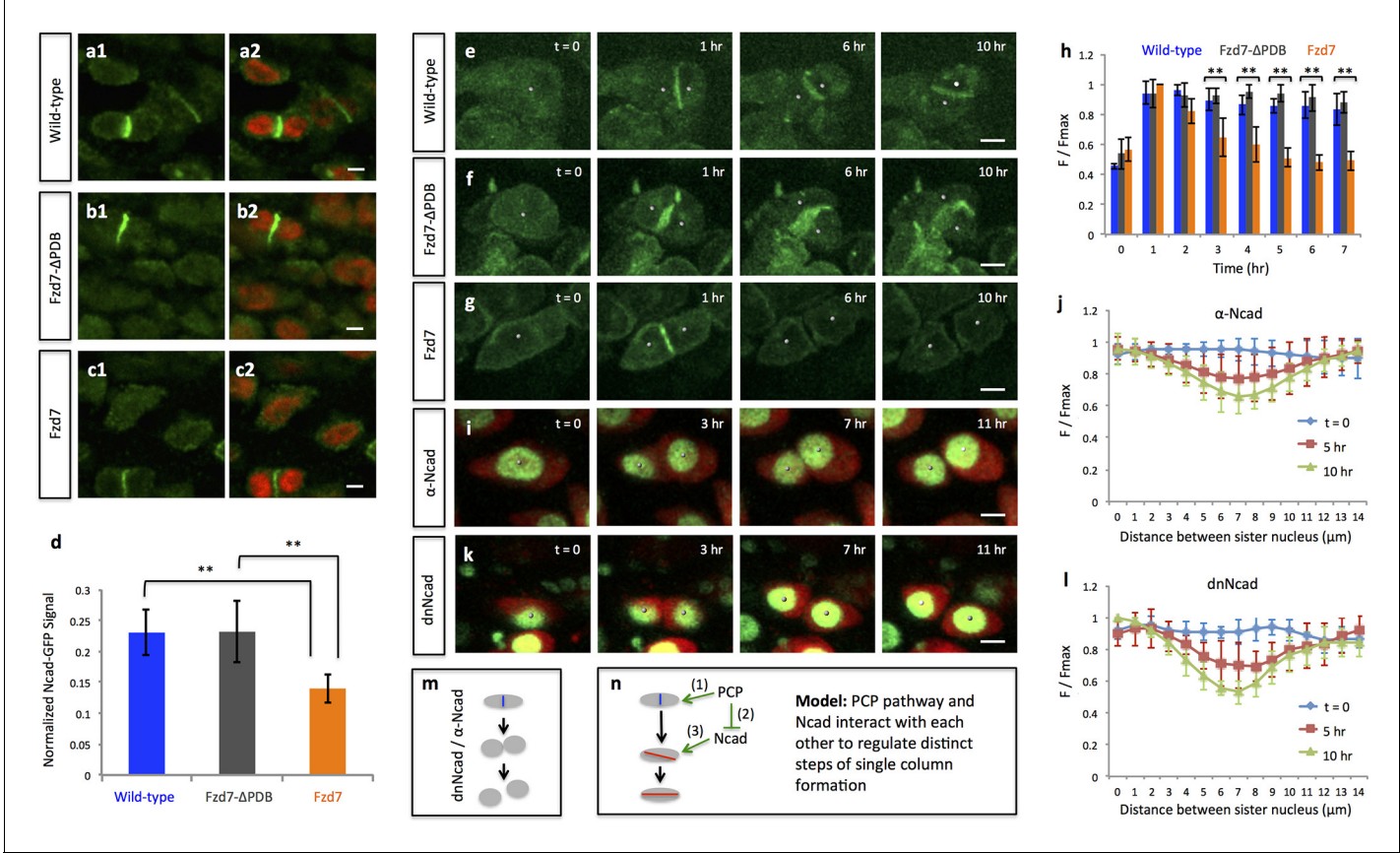

**Figure 6.** PCP signaling controls cell pivot by maintaining the local concentration of N-Cadherin. (**a–d**) Immunofluorescence with α-N-Cadherin (Ncad) antibody (green) and DAPI (red) in the frozen sections of wild-type tissues demonstrated that Ncad was present both on the membrane and in the cytoplasm, particularly enriched in cell-cell junctions (**a**). These patterns appeared to be unchanged in Fzd7-ΔPDB (**b**) or Fzd7 expressing cells (**c**). However, junctional Ncad in Fzd7 positive tissues was reduced (**c**). This was confirmed by normalizing junctional Ncad intensity to total Ncad intensity in each pair of sister cells (**d**) (n = 10, 10, eight for wild-type, Fzd7-ΔPDB, and Fzd7 tissues, respectively). See also *Figure 6—source data 1*. (**e–g**) In the chick metacarpals expressing Ncad-GFP (green) via RCAS infection, live imaging revealed that the fusion protein was enriched in the interface of sister cells in wild-type tissues (**e**) (n = 6) and tissues coexpressing Fzd7-ΔPDB (n = 6) over time (**f**). In contrast, in the presence of Fzd7, Ncad was initially concentrated in the junctions but then diminished as the cytoplasmic bridge between sister cells disconnected (**g**) (n = 7). (**h**) Quantifying Ncad-GFP expression along the post-cleavage furrow. GFP intensity was measured every one hour after cell division (T = 1 hr was the time when junctional Ncad-GFP was started to be observed) and normalized by dividing the maximal intensity during the time course (F/Fmax). Changes of cumulative F/Fmax showed that Ncad-GFP signal in Fzd7 expressing cells dropped about 50% 5 hr after cytokinesis. See also *Figure 6—source data 2*. (**i–l**) The normal function of Ncad is required for maintaining sister cell contact. Live imaging was performed on H2B-GFP (green) and mCherry (red) positive tissues cultured in the medium with α-Ncad antibody (1:10 dilution) (**i**), demonstrating the disengagement of sister cells (**j**) (n = 5). Similar cell behaviors were observed in the tissues expressing H2B-GFP (green) and a dominant-negative mutant of Ncad fused to mCherry through T2A sequence (red) (**k, l**) (n = 5). The signal intensity of the images in i was adjusted from the corresponding movie to present nuclei morphologies more clearly. In both cases, no complete pivoting was observed in cell doublets (n = 15 and 20 for α-Ncad antibody and dnNcad-T2A-mCherry, respectively). See also *Figure 6— source data 3*, *Figure 6—source data 4*. (**m**) Schematic diagram to show functional blocking of Ncad causes sister cells to separate. (**n**) Schematic diagram to summarize the roles of PCP signaling and junctional Ncad in regulating single column formation: (1) the absolute level PCP signaling is essential for oriented cell division, (2) PCP signaling reduces Ncad enrichment at the post-cleavage furrow, (3) normal junctional Ncad function is required for cell association and pivot. Scale bars: 4 μm. ** denotes p<0.01; * denotes p<0.05 (Wilcoxon Rank-Sum Test).

DOI: https://doi.org/10.7554/eLife.23279.042

The following source data and figure supplements are available for figure 6:

**Source data 1.** Fluorescence intensity measurement of endogenous junctional Ncad in wild-type tissues.
DOI: https://doi.org/10.7554/eLife.23279.050

**Source data 2.** Fluorescence intensity measurement of junctional Ncad-GFP in wild-type, Fzd7-ΔPDB or Fzd7 expressing tissues.
DOI: https://doi.org/10.7554/eLife.23279.051

**Source data 3.** Quantitative analysis of sister cell contact in the tissues treated with α-Ncad antibody.
DOI: https://doi.org/10.7554/eLife.23279.052

**Source data 4.** Quantitative analysis of sister cell contact in dnNcad expressing tissues.

*Figure 6 continued on next page*

*Figure 6 continued*

DOI: https://doi.org/10.7554/eLife.23279.053

**Figure supplement 1.** Ncad is enriched in the post-cleavage furrow of dividing cells.

DOI: https://doi.org/10.7554/eLife.23279.043

**Figure supplement 1—source data 1.** Pearson correlation analysis of junctional Ncad and phalloidin signal in wild-type tissues.

DOI: https://doi.org/10.7554/eLife.23279.046

**Figure supplement 2.** Perturbing PCP activity does not affect Ncad transcription.

DOI: https://doi.org/10.7554/eLife.23279.044

**Figure supplement 3.** Ncad-GFP intensity analysis confirms sister cell association.

DOI: https://doi.org/10.7554/eLife.23279.045

**Figure supplement 3—source data 1.** Cell-cell contact analysis in Ncad-GFP expressing tissues.

DOI: https://doi.org/10.7554/eLife.23279.047

**Figure supplement 3—source data 2.** Cell-cell contact analysis in Ncad-GFP and Fzd7-ΔPDB expressing tissues.

DOI: https://doi.org/10.7554/eLife.23279.048

**Figure supplement 3—source data 3.** Cell-cell contact analysis in Ncad-GFP and Fzd7 expressing tissues.

DOI: https://doi.org/10.7554/eLife.23279.049

# Materials and methods

## Plasmid construction

RIA and RCAS viral vectors were modified by introducing unique AscI and NotI digestion sites to facilitate cloning. For clonal analyses, CFP, GFP, membrane-GFP, H2B-YFP, mCherry, H2B-mCherry, Fzd7-ΔPDB, Fzd7, Fzd7-ΔPDB-T2A-CFP, Fzd7-ΔPDB-T2A-mCherry, Fzd7-ΔPDB-T2A-YFP, Fzd7-T2A-CFP, Fzd7-T2A-mCherry, Fzd7-T2A-YFP were cloned into RIA vector. For cell tagging, protein tagging and perturbation studies, H2B-GFP-T2A-mCherry, H2B-YFP, Ncad-GFP, dnNcad-T2A-mCherry, Fzd7-YFP, Fzd7-ΔPDB-YFP, Fzd7-YFP-VTTE, Fzd7-ΔVTTE-YFP were cloned into RCAS vector. The DNA sequences encoding KTxxxW and VTTE motif are located in the 1633–1650 and 1684–1701 nucleotides of chick Fzd7 gene (NM_204221.2), respectively. In RCAS-Fzd7-ΔKTxxxW-YFP and RCAS-Fzd7-ΔVTTE-YFP, the sequences encoding KTxxxW and VTTE were deleted, respectively. To clone RCAS-Fzd-YFP-VTTE, the DNA sequence encoding VTTE motif was added to the C terminus of YFP. dnNcad were previously described (*Li and Dudley, 2009*; *Packard et al., 2013*), and subcloned into the upstream of T2A-mCherry in RCAS vector. RCAS-Fzd7-ΔPDB, RCAS-DVL2-ΔPDZ and RCAS-Vangl2 were previously published (*Li and Dudley, 2009*).

## Viral concentration and infection

Recombinant RCAS plasmids were transfected into chick DF1 cells (ATCC, Manassas, VA; #CRL-12203, Lot number 62712171, Certificate of Analysis with negative mycoplasma testing available at ATCC website) in 10 cm culture dishes using standard transfection protocol. The transfected cells were further maintained in 15 cm dishes. When the cells were confluent, the cell culture medium was harvested once per day for three days, and was concentrated at 26,000 rpm for 1.5 hr. The pellet was dissolved in minimal volume of DMEM.

Recombinant RIA plasmids were cotransfected with Envelop A plasmid into DF1 cells in

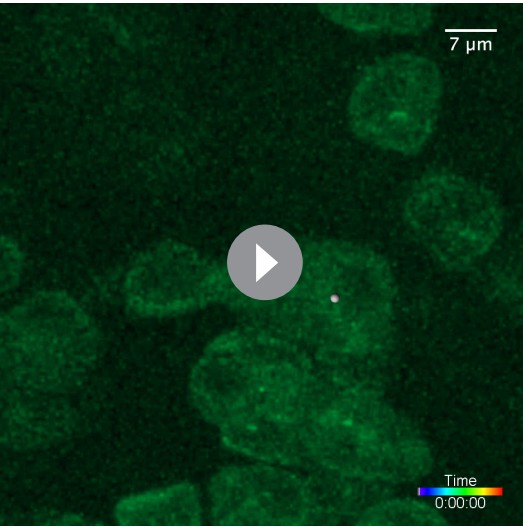

**Video 10.** Live imaging on the chick metacarpal expressing Ncad-GFP (green). The fusion protein was enriched in the post-cleavage furrow after cytokinesis and during the whole process of cell pivot. The snapshots of the segmented cells (white dots) were presented in *Figure 6e*. Scale bar: 7 μm

DOI: https://doi.org/10.7554/eLife.23279.054

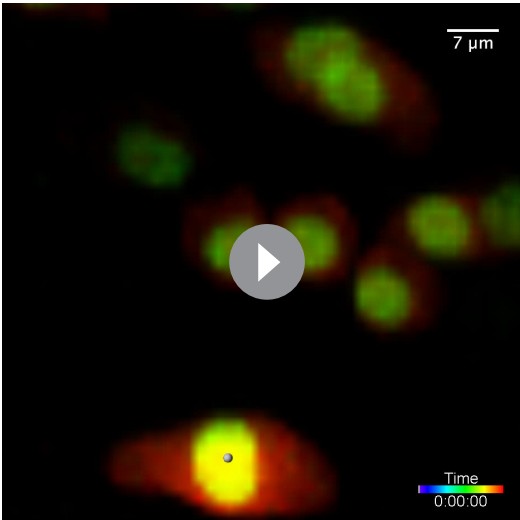

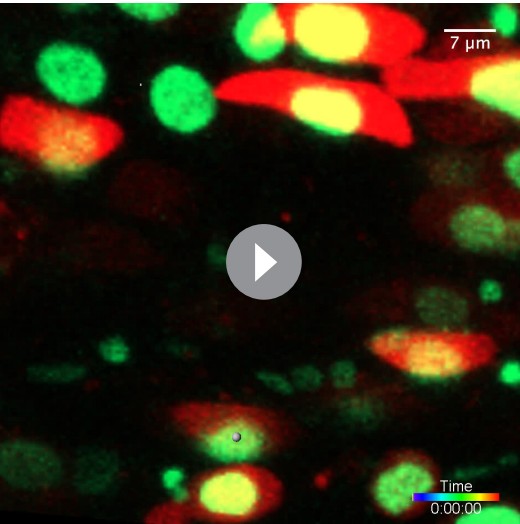

**Video 11.** Live imaging of sister cell separation in the H2B-GFP (green) and mCherry (red) expressing cartilage incubated in the medium containing an antibody against Ncad. The snapshots of the segmented cells (white dots) were presented in *Figure 6i*. Scale bar: 7 µm
DOI: https://doi.org/10.7554/eLife.23279.055

**Video 12.** Live imaging of sister cell separation in the chick metacarpal expressing H2B-YFP (green) and a dominant-negative mutant of Ncad (dnNcad) fused to mCherry with T2A sequence. The snapshots of the segmented cells (white dots) were presented in *Figure 6k*. Scale bar: 7 µm
DOI: https://doi.org/10.7554/eLife.23279.056

10 cm dishes. 24 hr later, the cell culture medium was harvested once per day for three days, and was concentrated using the same method as RCAS virus.

Both RCAS and RIA viruses were injected into chicken (Specific Pathogen-Free chicken, Charles River) right forelimbs at E3 (HH 19–20). The right humerus or metacarpal was dissected at E8 (HH 32–33) for further analysis.

## Immunofluorescence

Chick humerus was fixed in 4% PFA at 4°C for 30 min. Frozen tissues sections were permeabilized with blocking buffer (1xPBS with: 10% vol/vol normal goat serum, 1% BSA, 0.1% vol/vol Triton-X100, 0.025% sodium azide), stained with primary antibody (1:10 dilution for Rat anti-Ncad, DSHB) and then secondary antibody Goat anti-Rat Alexa-647 (1:500 dilution, Molecular Probes).

For double in situ hybridization, hybridization chain reactions (HCR) protocol was employed (*Choi et al., 2010*). Briefly, fixed frozen tissue sections were incubated with to anti-chick-Ncad and anti-chick-Fzd7 DNA probes for 16 hr at 45 degrees. The samples were further hybridized to the hairpins that contain both gene specific sequences and fluorophores at room temperature for signal amplification.

## Live imaging of cartilage explants and quantitative analysis of cell behaviors

The right metacarpal of chick embryos at E8 was dissected for organ culture. Molten agarose was

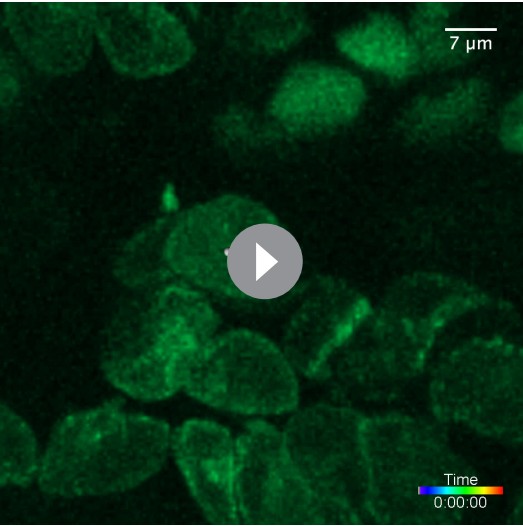

**Video 13.** Live imaging on the chick metacarpal expressing Ncad-GFP (green) and Fzd7-ΔPDB. Ncad-GFP was concentrated between sister cells during their rearrangement. The snapshots of the segmented cells (white dots) were presented in *Figure 6f*. Scale bar: 7 µm
DOI: https://doi.org/10.7554/eLife.23279.057

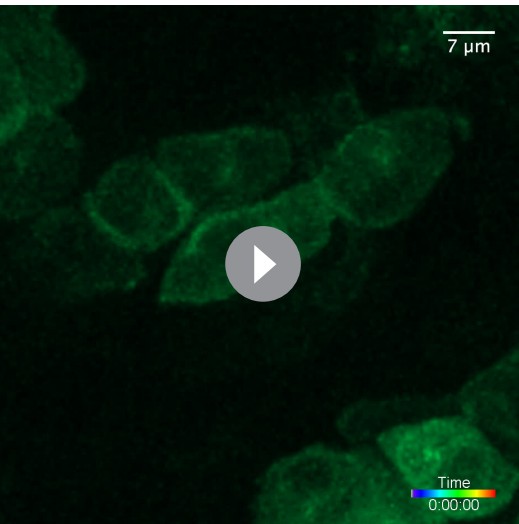

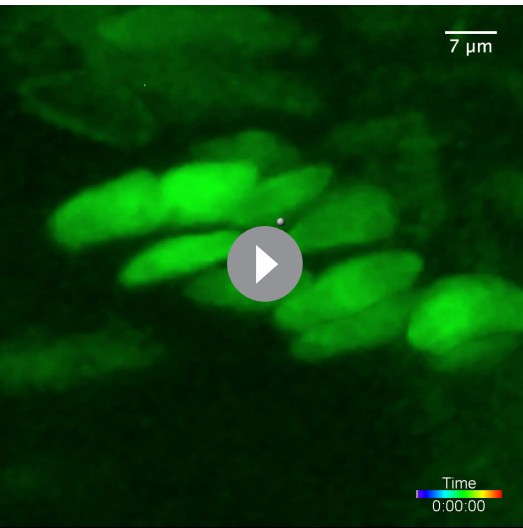

**Video 14.** Live imaging on the chick metacarpal expressing Ncad-GFP (green) and Fzd7. Ncad-GFP was initially concentrated at cell-cell contact and subsequently reduced as sister cells separated. The snapshots of the segmented cells (white dots) were presented in *Figure 6g*. Scale bar: 7 μm
DOI: https://doi.org/10.7554/eLife.23279.058

**Video 15.** Live imaging on the chick metacarpal expressing membrane-GFP (green) and GFP (green) via RIA infection. Intercalation occurred inside a GFP clone. The snapshots of the segmented cells (white dots) were presented in *Figure 2—figure supplement 2a*. Scale bar: 7 μm
DOI: https://doi.org/10.7554/eLife.23279.059

poured into the fluorodish (World Precision Instruments) and the custom-designed mold was immediately inserted into it. When the agarose was solidified, the mold was pulled out, leaving grooves in the agarose for holding the metacarpals. The metacarpals were submerged in DMEM/F12 growth

medium containing 0.2% bovine serum albumin, 50 mM ascorbate acid (Sigma-Aldrich, St. Louis, MO, USA), 10 mM glycerophosphate and 1% glutamine-penicillin-streptomycin (Invitrogen) in a humidified chamber at 37°C on the stage of the inverted laser scanning microscope (LSM 800 inverted, Carl-Zeiss) for live imaging.

For imaging chick metacarpal expressing H2B-GFP and mCherry, or the combination of H2B-YFP and dnNcad-T2A-mCherry, one-photon laser excitation was used with 0.6% and 0.8% relative power at wavelength of 488 and 561 nm, respectively; for imaging Ncad-GFP, 0.8% relative power at wavelength of 488 nm was used. In all the cases, optical sectioning was achieved at intervals of 1 μm and images were captured very one hour for 12 to 16 hr.

The images were imported into IMARIS 7.6.4 for cell morphology and trajectory analyses as previously described (*Li et al., 2015*). For cell pivot analysis, the x, y coordinates of sister cells were acquired over time. The pivot angle is an inverse tangent function of the distance between the two cells along y and x axis. The angle at time 0 is normalized to 0° for the ease of comparison. The direction parallel to the

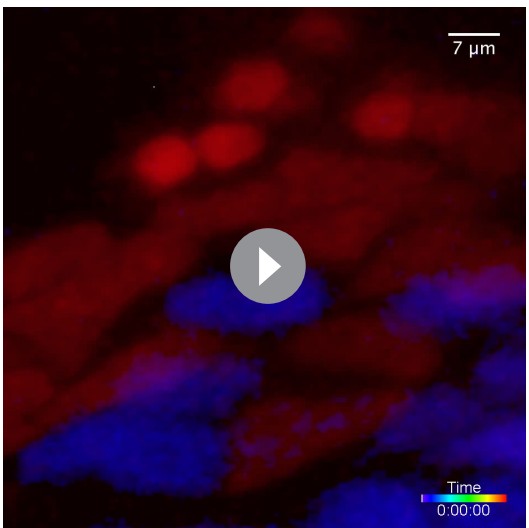

**Video 16.** Live imaging on the chick metacarpal expressing CFP (blue), mCherry (red) and H2B-mCherry (red) via RIA infection. Intercalation occurred between CFP positive and mCherry positive clones. The snapshots of the segmented cells (white dots) were presented in *Figure 2—figure supplement 2b*. Scale bar: 7 μm
DOI: https://doi.org/10.7554/eLife.23279.060

mediolateral axis of the bone is set to be 0°. All the angles were calculated using MATLAB.

## Measurement of fluorescence intensity

Measurement of fluorescence intensity in live and fixed samples with different region-of-interest shapes (including polyline kymograph analysis) was conducted in Image J.

## Statistical analysis

One-sample Kolmogorov-Smirnov test was used to assess the distribution of the datasets. If the datasets were not normally distributed, Wilcoxon Rank-Sum test was used for examining whether the differences between datasets were significant. Watson's $U^2$ test was used for evaluation the significance of orientation differences. It is also a nonparametric test that does not require the dataset to be normally distributed.

# Acknowledgements

This project is supported by DE024157 to MEB We thank Professor Carlos Lois, Professor Cheng-Ming Chuong and Caltech Biological Imaging Facility for sharing equipment.

# Additional information

### Competing interests

Marianne Bronner: Senior Editor, eLife. The other authors declare that no competing interests exist.

### Funding

| Funder | Grant reference number | Author |
| --- | --- | --- |
| National Institutes of Health | DE024157 | Marianne Bronner |

The funders had no role in study design, data collection and interpretation, or the decision to submit the work for publication.

### Author contributions

Yuwei Li, Conceived and designed the project, Constructed reagents and performed experiments, Performed quantitative analyses, Wrote the manuscript; Ang Li, Performed quantitative and statistical analysis, Helpful discussion; Jason Junge, Performed bioinformatic analyses of Frizzled-7 and Dishevelled-2 proteins, Helpful discussion; Marianne Bronner, Funding acquisition, Conceived and designed the project, Wrote the manuscript

### Author ORCIDs

Yuwei Li http://orcid.org/0000-0001-7753-4869
Marianne Bronner http://orcid.org/0000-0003-4274-1862

### Decision letter and Author response

Decision letter https://doi.org/10.7554/eLife.23279.061
Author response https://doi.org/10.7554/eLife.23279.062

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
