## [Decision Letter]

[Editors’ note: this article was originally rejected after discussions between the reviewers, but the authors were invited to resubmit after an appeal against the decision.]

Thank you for submitting your work entitled "PCP signaling coordinates oriented cell division and cell rearrangement in clonally expanding growth plate cartilage" for consideration by *eLife*. Your article has been reviewed by three peer reviewers, and the evaluation has been overseen by a Reviewing Editor and a Senior Editor. The reviewers have opted to remain anonymous.

Our decision has been reached after consultation between the reviewers. Based on these discussions and the individual reviews below, we regret to inform you that your work will not be considered further for publication in *eLife*.

As can be seen from the reviews the reviewers considered that this was an interesting paper addressing a key aspect of tissue morphogenesis that could, in principle, be suitable for publication in *eLife*. However all three pointed to a lack of suitable quantification of cell behaviours. There is also concern raised about how intercalation can be distinguished from cell rotation and how this was deduced from the videos and stills. Finally reviewer 1 felt that the use of the Fzd7 PDZ mutant was inappropriate and suggested that better PCP inhibitors should be employed, while reviewer 2 suggested the use of additional markers of the cytokinetic furrow would be necessary for accurate assessment of division planes.

While the large number of outstanding issues raised by the referees preclude simply a revision of this version of the paper, we would be happy to consider a new version of the manuscript containing further experiments that address the important points raised during the review process.

Reviewer #1:

This manuscript analyses morphogenesis of plate cartilage during limb development. The authors use cell labelling to follow the behaviour of cells in order to explain elongation across the proximal-distal axis. They found that clones of cells are reorganized in columns based in two mechanisms: oriented cell division and cell rearrangements. In addition they explore the role of PCP signalling and N-cadherin in these process.

Morphogenesis is a very important topic in general, and the study of orientated cell division and cell rearrangements has become increasingly central to understand development. However, the current manuscript contains too many problems as to be publishable in a journal like *eLife*. First, some of the conclusions are not supported by the data; second, some important experiments lack a proper quantitative analysis; third, some of the tools used to inhibit PCP signalling or N-cadherin are not properly justified.

Specific comments:

1) The authors claim that the clonal labelling shows complex columns along the PD axis. This is not obvious to me and more convincing data, together with proper quantification, is required

2) The data concerning cell intercalation (Figure 2, and Video 3 and Video 4) are not convincing; the intercalation is minimal, probably because the video is too short to see a more complete intercalation.

3) The authors claim that "cell rotation happens between sister cells, while cell intercalation takes place mainly between cousin cells", but they do not show any data to support this claim. The videos/images included in the manuscript are from different cells undergoing division, rotation or intercalation. In order to show that rotation is happing in sister cells, they need to show in the same time-lapse video that a dividing cell is followed by rotation.

4) The authors use a Fzd7 lacking the PDZ binding domain to inhibit PCP signaling. Fzd7 is the receptor for many Wnt signaling molecules, not all of them being PCP related. It is not clear how the expression of this truncated receptor could inhibit the endogenous Fzd7 activity or be a specific PCP inhibitor. Furthermore, the authors show that this mutated receptor is not localized in the membrane, where the endogenous Fzd7 receptor is expected to be localized. Better PCP signalling inhibitors should be used.

5) The authors study the localization of different Fzd7 mutants by comparing it with phalloidin staining. Fzd7 is a trans-membrane protein and therefore its localization should be compared with a membrane marker and not with cortical actin. In addition PCP signalling is known to affect the cytoskeleton and it is possible that some of the mutants could affect the distribution of actin. Indeed phalloidin staining looks quite different between c2/d2 and b2/e2.

6) EGTA is a strong treatment, is it reversible? How do the authors know that is affecting only N-cadherin? There is no quantification of this treatment or the N-cad antibody treatment. In addition, there are several dominant negatives of N-cadherin that could be used to strength the conclusion that cell rotation requires N-cad, which is rather weak as it is currently presented

7) The authors use Figure 5 to conclude that Fzd7 overexpression inhibits N-cad localization at the junction between sister cells. However I can clearly see a doublet of cells expressing N-cad at the junction in the bottom of the picture. The other cells shown in Figure 5 that are negative for N-cadherin are not even in contact. These data do not support the author conclusion.

8) The authors mention that reduction of N-cadherin between sister cells induced by Fz7 leads to cell separation and they refer to Figure 5 and Video 14, but I cannot see any cell separation at all in this figure or video

Reviewer #2:

This interesting paper applies novel labeling and imaging strategies to advance our understanding of cell behaviors driving elongation of long bones. In principle, it could be an excellent contribution to an important field. However, the paper suffers from a variety of nagging issues with the data. If more careful quantification of the results confirms the claims made here and more care is placed into the writing, the paper should be acceptable for *eLife*.

Major comments:

Subsection “Oriented cell division and cell rotation are interconnected but differentially regulated by PCP signaling”. Claims about sister cell separation are made, but no quantification is supplied. Distance between new sisters over time would provide a simple metric for this important observation.

In subsection “Influence of PCP signaling on cell rotation involves local enrichment of N-Cadherin in the cell division plane”. The claim is made that N-cad labels the midbody, but surely this is not correct. Also, the definition provided for the midbody (subcellular structure in the midpoint between two future sister) is not accurate. The midbody is formed as the microtubules of the mitotic spindle are bundled together by the cytokinetic furrow. Without tubullin staining this claim cannot be made. Also, I think that not being able to visualize the actual cell membrane during division and "rotation" makes it hard to say what is going on here. What appears to be labelled by N-cad is the cytokinetic furrow, which would also be marked by phospho-myosin or active RhoA. Given the key role of PCP proteins in governing actomyosin contraction, the authors would be well served by examing such markers of the furrow here.

Figure 5. What happens to N-cad localization during the process they authors call "rotation" This seems to be an important omission here.

In the same section no numbers are provided to back the claim that rotation fails, though this metric has been quantified in other parts of the paper.

As above, separation should be quantified.

Also in that section the level of N-cad at the "midbody" (more likely the furrow, see above) is not quantified, but must be to support this claim.

Reviewer #3:

This paper begins by using clonal analysis to examine cell rearrangements in the growth plate cartilage of the chicken limb. Doing this, they follow two types of cell rearrangements, cell rotation and mediolateral intercalation. They put a lot of emphasis on how they "expand the toolkit for high-resolution clonal analysis in a non-genetic system"; however, this approach seems unsurprising and not particularly novel. Furthermore, it has been thought for many years that chondrocyte columns arise clonally by oriented division followed by cell rearrangement. However, I do think this is a nice, simple system to study the problem and while similar studies have been performed they are arguably too simple (eg zebrafish Meckel's cartilage) or too complex to interpret and not accessible (eg mouse cartilage).

First, because the authors set up the paper by introducing their clonal analysis, I expected more detail on single vs complex columns of chondrocytes, perhaps with better definitions and quantitation. These seem like simple concepts, but are not made clear in the text. They state that "cartilage is a mosaic of simple and complex monoclones, with a minor contribution from intermingled clones" What are the criteria for "simple" versus "complex"? How do they distinguish between "complex" columns versus columns that have not finished intercalating? Can you distinguish between misorientation and cell shape change? Also, it seems to me you need more than two cells in order to undergo intercalation (subsection “Complex column formation involves mediolateral cell intercalation”).

Second, I was a bit disappointed that the authors did not continue their clonal analyses once they moved into manipulating the PCP pathway. The prediction would have been that by perturbing the PCP pathway, the clones would be unable to reorient or intercalate, which might be easily seen in the clonal analysis. However, instead of going back to the clonal analyses, the authors move toward examining single cell division orientation over a (comparatively) brief period of time. How do we know that these neighbor relationships are now stable (or even if they should remain stable)? So, while I found the Fzd mutational analysis interesting, it seems to me that these experiments don't really address the initial questions and really are much more focused on the immediate effects of division to sister cells, rather than the clonal expansion of the growth plate.

Many of the Materials and methods are not very clear. We need more detail on the constructs. For example, in subsection “Excess membrane-bound Frizzled-7 receptor inhibits cell rotation” it states that adding -VTTE motif promotes membrane localization – presumably this is not the WT construct. Does this mean they put the VTTE c-terminal to the YFP?

EGTA experiment (presumably some kind of control?) is not explained.

The videos do not add much for me as there are no additional intervening frames compared to what is shown in the stills, and I find it difficult to interpret the rotation or intercalation, especially since we have very little sense of the greater context. It would be helpful to see more of a tissue-level snapshot to see the trend overall, rather than individual cells. It would also be much more satisfying to see the clonal analysis over time. (Is it possible to culture these limbs?).

[Editors’ note: what now follows is the decision letter after the authors submitted for further consideration.]

Thank you for resubmitting your work entitled "Planar cell polarity signaling coordinates oriented cell division and cell rearrangement in clonally expanding growth plate cartilage" for consideration by *eLife*. Your article has been re-reviewed by three peer reviewers, and the evaluation has been overseen by a Reviewing Editor and a Senior Editor. Unfortunately all three of the original reviewers, while appreciating that the paper is much improved, still felt that key concerns remain un-addressed in the revised version. These have been clearly enunciated in the three sets of comments.

In particular the reviewers highlight the need for a membrane label to more clearly monitor the cell behaviours, as well as the live imaging of the differently coloured clones which are considered insufficiently convincing to discriminate intercalation versus cell pivoting by cells in these columns. Hopefully you'll find the reviewers comments helpful in performing further revisions of the work.

Reviewer #1:

This is an improved version of the manuscript and the authors have addressed most of my previous comments. I am still not completely convinced about the quantification shown in Figure 1, as it is not evident from these data how the distribution of frequencies of simple and complex columns is. However this is now a minor point.

Reviewer #2:

The authors make use of clonal analysis to follow cellular rearrangements during elongation of the growth plate cartilage in the chicken limb. The initial submission was flawed due to a lack of precision in the original analysis (in defining terms, quantification of data, etc) and in difficulties with seeing and interpreting the relevant cell movements in the data/videos. The writing in this revision is much improved – clearer and more precisely written than in the original submission. Substantial quantitation has been added to the paper. However, I still find myself struggling to see the data that the authors highlight.

A major issue in the previous version was the distinction between single and complex columns. I'm still not sure of this: I think Figure 1F1 shows a single column (in red) and the text states that Figure 1F2 shows complex columns in "blue and green clones". Then, the authors go on to say the "a minority of complex ones (8%) intermixed with non-clonally related cells (white arrow pointing to the uninfected cell beside the green clone, Figure 1F1)". What is the justification for calling this a complex column? Don't they have any better examples of mixed color complex columns?

I appreciate the difficulties with live imaging of different colored cell clones; however, the videos are still unsatisfying and are not very definitive with regards to intercalation or "cell-pivoting". The "zoomed-out" view certainly helps. But, the authors argue that they cannot perform live imaging due to the limitations of photo-damage on the tissues. In fact, I think it would have been better to skip the live imaging and take live snapshots at several time points which would allow us to better follow the clones over multiple days.

The live imaging would be interesting if we could actually see the cell behaviour/morphology (e.g. something sort of hinted at in the pictures of Fzd7-membrane in Figure 5). Instead, we get none of the benefits of following the clones, and very little appreciation of the actual immediate cellular movements. Finally, doing the single color clone in the PCP perturbations is also not very satisfying, for the same reason above, as it is difficult to understand which columns are "complex stacks" and which are "disorganised".

I would disagree with their statement in their Response to Reviewers: "As mediolateral intercalation is well known to be important in cartilage development[…] we chose instead to focus[…]" I do not think this mediolateral intercalation is well-established, as the major evidence is from limited zebrafish Meckel's cartilage studies. The data in this paper still leaves me wondering: does the intercalation actually occur in this system?

Reviewer #3:

This revised manuscript is generally improved over the original, as efforts have been made to quantify the observed results.

However, concerns do remain. The quantification of cytoplasmic FP intensity does allow the authors say a bit more about the data they obtained, but this is not really the correct fix. As per my original review, the absence of a membrane label is still problematic as it impedes a clear understanding of what is really going on with these cells. Given the complexity of the authors' claims, this remains a considerable issue.

In addition, quantification of raw pixel intensity is to a useful metric of N-cad levels for Figure 6. Typically, this value would be normalized in some way (to a membrane label, or to total signal).

---

## [Author Response]

[Editors’ note: the author responses to the first round of peer review follow.]

Reviewer #1:This manuscript analyses morphogenesis of plate cartilage during limb development. The authors use cell labelling to follow the behaviour of cells in order to explain elongation across the proximal-distal axis. They found that clones of cells are reorganized in columns based in two mechanisms: oriented cell division and cell rearrangements. In addition they explore the role of PCP signalling and N-cadherin in these process.Morphogenesis is a very important topic in general, and the study of orientated cell division and cell rearrangements has become increasingly central to understand development. However, the current manuscript contains too many problems as to be publishable in a journal like eLife. First, some of the conclusions are not supported by the data; second, some important experiments lack a proper quantitative analysis; third, some of the tools used to inhibit PCP signalling or N-cadherin are not properly justified.

We have softened the conclusions to be more circumspect, have improved the quantitation and have improved descriptions to better explain the tools.

Specific comments:1) The authors claim that the clonal labelling shows complex columns along the PD axis. This is not obvious to me and more convincing data, together with proper quantification, is required

We thank the reviewer for pointing out that these data were not sufficiently convincing. To address this important point, we have now presented new representative images to show the morphologies of both single and complex columns (Figure 1F1, 1F2). Furthermore, we have performed quantitative analysis to distinguish single from complex columns as well as cells that are not aligned (Figure 1). We also have added quantitation to measure column orientation (Figure 1). The results show that while single columns extended along the tissue proximodistal axis, complex columns were slightly shifted (Figure 1), but still organized in contrast to the arbitrary arrangement of clones in the resting zone (Figure 1H2) (Figure 1—figure supplement 1). Taken together, this strongly suggests that proliferative but not resting clones are polarized.

2) The data concerning cell intercalation (Figure 2, and Video 3 and Video 4) are not convincing; the intercalation is minimal, probably because the video is too short to see a more complete intercalation.

The reviewers point is well taken. In fact, cell intercalation in growing cartilage is well known to occur during cartilage formation (1, 2). The previous live imaging to follow this long-term process (more than 48 hors) was performed on GFP expressing cartilage using two-photon laser to avoid tissue damage (34). However, here we have to use one-photon laser to track the behaviors of multicolored clones. Unfortunately, this has some inherent limitations with respect to depth of penetration and length of imaging (24 hours or less for cartilage explants). Given these constraints, our results confirm that partial intercalation occurs more frequently between clonally related cells (Figure 2) (Supplementary Video 4) (Supplementary Video 5); however, the 24 hr time period is not sufficient to catch the complete cell intercalation process. We now downplay cell intercalation, which was already well known, and only mention these results briefly as it is a minor portion of the paper.

In the current version of the manuscript, our aim was to determine the mechanism regulating single column formation. Unlike mediolateral intercalation for complex columns, the mechanism underlying single column formation was not well understood. Therefore, our perturbation studies were focused on the molecular mechanisms driving cell pivot (previously referred to as “cell rotation” in the original manuscript). We have now reorganized the manuscript to focus on our novel results showing the importance of cell pivot to generate simple columns. Our perturbation experiments then go on to test the role PCP signaling and N-cadherin in this pivot process.

3) The authors claim that "cell rotation happens between sister cells, while cell intercalation takes place mainly between cousin cells", but they do not show any data to support this claim. The videos/images included in the manuscript are from different cells undergoing division, rotation or intercalation. In order to show that rotation is happing in sister cells, they need to show in the same time-lapse video that a dividing cell is followed by rotation.

The reviewer raises an important and valid point. To address this, we have performed additional live imaging and now show a better example of a sister starting to pivot following cell division (Figure 2) (Supplementary Video 2). Moreover, we show that chondrocytes that express NcadGFP undergo cell division followed by cell pivot into columns (Figure 6) (Supplementary Video 12). These combined results support the model of cell pivot between sister cells.

4) The authors use a Fzd7 lacking the PDZ binding domain to inhibit PCP signaling. Fzd7 is the receptor for many Wnt signaling molecules, not all of them being PCP related. It is not clear how the expression of this truncated receptor could inhibit the endogenous Fzd7 activity or be a specific PCP inhibitor. Furthermore, the authors show that this mutated receptor is not localized in the membrane, where the endogenous Fzd7 receptor is expected to be localized. Better PCP signalling inhibitors should be used.

The reviewer raises a good point. To address the issue of specificity, we have performed a secondary means of knock-down. To this end, we have employed a dominant-negative mutant of Dishevelled which lacks the PDZ domain (DVL-ΔPDZ) such that it strongly inhibits the PCP pathway with little or no effect on the canonical Wnt pathway (3, 4, 5). When we express this reagent in chick cartilage, we find that its effects on cell behavior are similar to the dominant-negative mutant of Fzd7 (Figure 3) (Supplementary Video 8) (Supplementary Video 9). Finally, we reference previous literature showing that perturbation of β-catenin in both chick and mouse cartilage suggests a function in chondrocyte proliferation and differentiation, but no role in oriented cell division or cell morphology (6, 7). Taken together, these results support a role for the PCP pathway rather than canonical Wnt signaling in chondrocyte column formation.

5) The authors study the localization of different Fzd7 mutants by comparing it with phalloidin staining. Fzd7 is a trans-membrane protein and therefore its localization should be compared with a membrane marker and not with cortical actin. In addition PCP signalling is known to affect the cytoskeleton and it is possible that some of the mutants could affect the distribution of actin. Indeed phalloidin staining looks quite different between c2/d2 and b2/e2.

To address this important point, we have now normalized the sub-cellular localization of different Fzd7 mutants to the membrane marker, membrane-RFP (Figure 5—figure supplement 1). The results are similar to that we previously found with phalloidin staining (Figure 5).

6) EGTA is a strong treatment, is it reversible? How do the authors know that is affecting only N-cadherin? There is no quantification of this treatment or the N-cad antibody treatment. In addition, there are several dominant negatives of N-cadherin that could be used to strength the conclusion that cell rotation requires N-cad, which is rather weak as it is currently presented.

We thank the reviewer for this suggestion and agree that EGTA has broad effect making it difficult to prove its specificity on N-cadherin activity. Hence, these data have been removed from the revised manuscript. We have used Ncad antibody since it has previously been shown to effectively block N-cadherin function in micromass cultures made from chick limb mesenchyme cells (8). We titrated the supernatant containing this antibody and found that cartilage explants incubated in the medium supplemented with a 1:10 dilution became disorganized (Supplementary Video 13). Therefore, we used this culture condition for imaging and quantitative analysis (Figure 6).

As a secondary means of knock-down, we have now added a previously established dominant-negative mutant of N-cadherin (9). The results show that sister cells ectopically expressing this protein become separated over time (Figure 6) (Supplementary Video 14), similar to effects with Ncad antibody. We thank the reviewer for suggesting the use of the dominant negative construct.

7) The authors use Figure 5 to conclude that Fzd7 overexpression inhibits N-cad localization at the junction between sister cells. However I can clearly see a doublet of cells expressing N-cad at the junction in the bottom of the picture. The other cells shown in Figure 5 that are negative for N-cadherin are not even in contact. These data do not support the author conclusion.

We apologize that this was not more clear. Comparing Figure 6 and Figure 6, the expression level of junctional Ncad is reduced in the cells expressing an intact Fzd7. To clarify this finding, we now quantitate this difference by performing fluorescence intensity analysis of junctional Ncad (Figure 6). These data are consistent with our model in which high PCP activity down-regulates the amount of junctional Ncad.

8) The authors mention that reduction of N-cadherin between sister cells induced by Fz7 leads to cell separation and they refer to Figure 5 and Video 14, but I cannot see any cell separation at all in this figure or video.

We thank the reviewer for pointing this out. To address this, we have now extended our imaging session to clearly show that sister cells are separated (Figure 6) (Supplementary Video 16).

Reviewer #2:This interesting paper applies novel labeling and imaging strategies to advance our understanding of cell behaviors driving elongation of long bones. In principle, it could be an excellent contribution to an important field. However, the paper suffers from a variety of nagging issues with the data. If more careful quantification of the results confirms the claims made here and more care is placed into the writing, the paper should be acceptable for eLife.Major comments:Subsection “Oriented cell division and cell rotation are interconnected but differentially regulated by PCP signaling”. Claims about sister cell separation are made, but no quantification is supplied. Distance between new sisters over time would provide a simple metric for this important observation.

We thank the reviewer for pointing this out. To address this, we have now measured the fluorescence intensity of the sister cells expressing cytoplasmic mCherry. If sister cells are separated after cytokinesis, mCherry intensity should significantly drop in the middle of the cells; otherwise, this value should remain more or less constant (Figure 2). The results show that wild-type cells (Figure 2), Fzd7-ΔPDB expressing cells (Figure 3) and DVL-ΔPDZ expressing cells (Figure 3) are associated; in contrast, Fzd7 expressing cells (Figure 3), Vangl2 expressing cells (Figure 3) and cells with inhibited Ncad functions (Figure 6) are separated. This supports the idea that inhibiting PCP signaling causes misoriented cell division, but does not impact sister cell association; in contrast, promoting PCP signaling or inhibiting Ncad activity results in sister cell separation.

In subsection “Influence of PCP signaling on cell rotation involves local enrichment of N-Cadherin in the cell division plane”. The claim is made that N-cad labels the midbody, but surely this is not correct. Also, the definition provided for the midbody (subcellular structure in the midpoint between two future sister) is not accurate. The midbody is formed as the microtubules of the mitotic spindle are bundled together by the cytokinetic furrow. Without tubullin staining this claim cannot be made. Also, I think that not being able to visualize the actual cell membrane during division and "rotation" makes it hard to say what is going on here. What appears to be labelled by N-cad is the cytokinetic furrow, which would also be marked by phospho-myosin or active RhoA. Given the key role of PCP proteins in governing actomyosin contraction, the authors would be well served by examing such markers of the furrow here.

We thank the reviewer for this important comment and agree that using the term midbody was a misnomer. Rather, we should have called it the cleavage furrow. To address the reviewer’s point, we first tried immunofluorescence experiment on chick cartilage using both phospho-myosin antibody and active RhoA antibody; however, neither of these gave a signal. Therefore, we chose to normalize junctional Ncad signal to phalloidin signal, which labels the contractile ring in the cleavage furrow. In most analyzed samples, junctional Ncad is not located in the cleavage furrow (Figure 6—figure supplement 1). On the other hand, our live imaging analysis on Ncad-GFP expressing cells clearly shows that Ncad is enriched between sister cells after division and during cell pivot (Figure 6). These combined data suggest that Ncad is enriched in the post-cleavage furrow and controls sister cell rearrangement.

Given the role of PCP proteins in governing actomyosin contraction and oriented cell division, an intriguing possibility is that PCP proteins recruit Ncad to the post-cleavage furrow to establish adherens junctions after cytokinesis and this adhesive interface is critical for subsequent cell rearrangement.

Figure 5. What happens to N-cad localization during the process they authors call "rotation" This seems to be an important omission here.

To address this interesting point, we have now performed new live imaging to show that junctional Ncad-GFP is enriched in both pivoting wild-type cells (Figure 6) (Supplementary Video 12) and Fzd7-ΔPDB expressing cells (Figure 6) (Supplementary Video 15).

In the same section no numbers are provided to back the claim that rotation fails, though this metric has been quantified in other parts of the paper.

To address this, we now provide numbers to support the conclusions.

As above, separation should be quantified.

As requested, we have now quantified cell separation in all conditions.

Also in that section the level of N-cad at the "midbody" (more likely the furrow, see above) is not quantified, but must be to support this claim.

As requested, we have now measured the fluorescence intensity of both endogenous Ncad (Figure 6) and ectopic Ncad-GFP (Figure 6) at cell-cell contacts. Our data support the idea that Ncad is concentrated in the post-cleavage furrow and subsequently maintained in the interface between sister cells.

Reviewer #3:This paper begins by using clonal analysis to examine cell rearrangements in the growth plate cartilage of the chicken limb. Doing this, they follow two types of cell rearrangements, cell rotation and mediolateral intercalation. They put a lot of emphasis on how they "expand the toolkit for high-resolution clonal analysis in a non-genetic system"; however, this approach seems unsurprising and not particularly novel. Furthermore, it has been thought for many years that chondrocyte columns arise clonally by oriented division followed by cell rearrangement. However, I do think this is a nice, simple system to study the problem and while similar studies have been performed they are arguably too simple (eg zebrafish Meckel's cartilage) or too complex to interpret and not accessible (eg mouse cartilage).First, because the authors set up the paper by introducing their clonal analysis, I expected more detail on single vs complex columns of chondrocytes, perhaps with better definitions and quantitation. These seem like simple concepts, but are not made clear in the text. They state that "cartilage is a mosaic of simple and complex monoclones, with a minor contribution from intermingled clones" What are the criteria for "simple" versus "complex"? How do they distinguish between "complex" columns versus columns that have not finished intercalating? Can you distinguish between misorientation and cell shape change? Also, it seems to me you need more than two cells in order to undergo intercalation (subsection “Complex column formation involves mediolateral cell intercalation”).

We thank the reviewer for bringing this confusion to our attention, which we hope to have rectified in the revised manuscript. By simple column, we mean columns comprised of clonally related cells that are one cell diameter in width. Complex columns are also mainly clonal but two or more cells wide. To address the frequency of these columns and distinguish between them, we have performed further quantitative analysis. This helps to distinguish columns from non column (Figure 1) and single column from complex column (Figure 1). We also measured the orientation of both single and complex columns (Figure 1). The data support the idea that chondrocytes in the proliferative zone form two types of columns: single and complex, both of which undergo polarized growth.

With respect to distinguishing between misorientation and cell shape change, we have now measured these two cellular parameters and shown that mutant cells are misoriented and morphologically more isotropic (Figure 3—figure supplement 1). With respect to the relationship between cell number and cell intercalation, we found that intercalation only occurs between more than two cells as the reviewer suggests.

An ideal method to distinguish "complex" columns versus columns that have not finished intercalating” would be to perform clonal analysis and live imaging on the same samples. However, the one-photo laser causes photo-damage to the tissues such that the maximal imaging session is 24 hours or less, which is not sufficient to observe the complete process of cell intercalation in cartilage (2). Therefore, we can only define simple and complex clones in tissues sections.

As mediolateral intercalation is well known to be important in cartilage development, we chose instead to focus the current manuscript on determining the mechanism whereby single column formation is regulated, as this is novel. Therefore, our functional experiments focused on perturbing single column formation. We have performed further functional experiments combining clonal analysis and functional perturbation as requested (see below).

Second, I was a bit disappointed that the authors did not continue their clonal analyses once they moved into manipulating the PCP pathway. The prediction would have been that by perturbing the PCP pathway, the clones would be unable to reorient or intercalate, which might be easily seen in the clonal analysis. However, instead of going back to the clonal analyses, the authors move toward examining single cell division orientation over a (comparatively) brief period of time. How do we know that these neighbor relationships are now stable (or even if they should remain stable)? So, while I found the Fzd mutational analysis interesting, it seems to me that these experiments don't really address the initial questions and really are much more focused on the immediate effects of division to sister cells, rather than the clonal expansion of the growth plate.

We thank the reviewer for this suggestion. To address this point, we have now combined clonal analysis with perturbation studies to observe clone morphologies in the tissues expressing either Fzd7-ΔPDB or Fzd7. The results show that many Fzd7-ΔPDB expressing clones form misoriented single and complex stacks (Figure 4); in contrast, Fzd7 expressing clones are comprised of truly disorganized cells (Figure 4). These data support a model in which distinct cellular behaviors (oriented division and cell rearrangement) are sensitive to the activity of PCP pathway (Figure 4).

Many of the Materials and methods are not very clear. We need more detail on the constructs. For example, in subsection “Excess membrane-bound Frizzled-7 receptor inhibits cell rotation” it states that adding -VTTE motif promotes membrane localization – presumably this is not the WT construct. Does this mean they put the VTTE c-terminal to the YFP?EGTA experiment (presumably some kind of control?) is not explained.

We apologize for this omission. To address this, we have now expanded and clarified the description of different mutations of this construct and other reagents and protocols (See Materials and methods for details).

We agree that EGTA lacks specificity so these data have been replaced by using an alternative approach for blocking N-cadherin activity. To this end, we employ a previously established dominant-negative mutant of N-cadherin (9) and show that sister cells ectopically expressing this protein become separated over time (Figure 6).

The videos do not add much for me as there are no additional intervening frames compared to what is shown in the stills, and I find it difficult to interpret the rotation or intercalation, especially since we have very little sense of the greater context. It would be helpful to see more of a tissue-level snapshot to see the trend overall, rather than individual cells. It would also be much more satisfying to see the clonal analysis over time. (Is it possible to culture these limbs?)

To address this, we now present the videos in a more “zoomed-out” view such that neighboring cells can also be followed. Unfortunately, we are unable to perform live imaging on the whole tissue because it requires using low magnification objective lenses, which do not provide sufficient spatial resolution to visualize cell pivot behavior.

With respect to following clonal analysis over time, we tried hard to perform live imaging on the explanted cartilage labeled with distinct fluorophores at clonal density, However, the one-photo laser causes photo-damage to the tissues such that the maximal imaging session is only up to 24 hours, which is not sufficient to observe the complete process. We did try to culture whole limb for live imaging, but to no avail as neighboring tissues such as tendon and perichondrium reduced light penetration.

[Editors’ note: the author responses to the re-review follow.]

Reviewer #1 and 2Reviewer 1 raised minor concerns and reviewer 2 has major concerns on our quantitative analysis of clone morphologies, especially how to assess complex columns.Reviewer 1: “I am still not completely convinced about the quantification shown in Figure 1, as it is not evident from these data how the distribution of frequencies of simple and complex columns is. However this is now a minor point”.Reviewer 2: “A major issue in the previous version was the distinction between single and complex columns. I'm still not sure of this: I think Figure 1F1 shows a single column (in red) and the text states that Figure 1F2 shows complex columns in "blue and green clones". Then, the authors go on to say the "a minority of complex ones (8%) intermixed with non-clonally related cells (white arrow pointing to the uninfected cell beside the green clone, Figure 1F1)". What is the justification for calling this a complex column? Don't they have any better examples of mixed color complex columns?”.

We present representative images (Figure 1), which are further supported by quantitative characterization of column versus non-column, single versus multi column, column orientation (Figure 1). While a single column (red in Figure 1F1) and a multi column (blue in Figure 1F2) are aligned along the tissue elongation axis, there is also a multi column with shifted orientation (green in Figure 1F2). These representative images are consistent with quantitative analysis of column orientation (Figure 1).

We agree that our quantitative analysis is not sophisticated enough to discriminate complex columns at potentially distinct dynamic states, i.e. whether they are stable, intercalating or disorganized ones. Hence, to more accurately describe clone morphologies, we rephrased “complex columns” into “multi-column clones”. Additionally, we better emphasize the novelty of our study and clarified that our study is focused on single column formation (Introduction section and subsection “Inhibiting PCP signaling disrupts oriented cell division, but not cell pivot behavior”). Such changes do not weaken our conclusion on the major part of the story.

Reviewer 2 proposed one experiment to characterize mediolateral intercalation. “I appreciate the difficulties with live imaging of different colored cell clones; however, the videos are still unsatisfying and are not very definitive with regards to intercalation or "cell-pivoting". The "zoomed-out" view certainly helps. But, the authors argue that they cannot perform live imaging due to the limitations of photo-damage on the tissues. In fact, I think it would have been better to skip the live imaging and take live snapshots at several time points which would allow us to better follow the clones over multiple days.”

Although we agree that such a “snapshot” experiment could provide novel insights into the dynamic process of multi column formation, unfortunately, it is not technically feasible given current technology. Explant cartilage cultures tend to drift and rotate during their growth. Further complicating this experiment, following live tissues using snapshots for several days would require changing culture media and constantly transferring explant culture from the incubator to the microscope stage and back.

These processes cause tissues to move and would make it impossible to follow the same clones. Rigorously performing such experiments requires the development of long-term limb culture. To our knowledge, there is no literature on maintaining normal tissue growth over such a long-term period. Moreover, characterizing the behavior underlying mediolateral intercalation will require two-photon live imaging and perturbation studies to clarify the cellular and molecular mechanisms. These experiments would warrant a full paper on their own and would take considerable time to accomplish. For these reasons, we have focused the current study on single column formation and cell pivot and the mechanisms underlying behavior.

Reviewer 2 doubts that live imaging deepens our understanding of cell rearrangement “The live imaging would be interesting if we could actually see the cell behaviour/morphology (e.g. something sort of hinted at in the pictures of Fzd7-membrane in Figure 5). Instead, we get none of the benefits of following the clones, and very little appreciation of the actual immediate cellular movements ".

We partially agree with the reviewer’s statement about using live imaging to follow clones with distinct colors. As we pointed out in our previous response letter, chondrocyte cell intercalation is a very slow process (2) and following it in the long-term will require more advanced imaging techniques than currently available. We appreciate that the reviewer suggests a snapshots experiment, but as discussed above, it has many technical challenges. For these reasons, we toned down our statement about cell intercalation and multi column formation (Introduction section, subsection “Multi-column clone formation appears to involve mediolateral cell intercalation” and Discussion paragraph one). Furthermore, we moved our observation of mediolateral intercalation into the end of the manuscript (Result section final paragraph), and also moved the live imaging snapshots of this cellular behavior into Supplementary Data (Figure 2—figure supplement 2). These rearrangements will focus readers’ attention on cell pivoting and single column formation.

We disagree with the reviewer’s general statement about the application of live imaging to studying chondrocyte behavior. Our work shows that live imaging provides a direct means to visualize the sequential steps of single column formation: oriented cell division, sister cell association and subsequent cell pivoting. Such a dynamic process could only be inferred from static studies. Ours is the first work that definitively explains this process in limb skeleton. Importantly, we quantitatively demonstrate that cell pivoting is a not a classical mediolateral intercalation process, as reviewer 3 has pointed out in his previous comments (“To be clear, this behavior between new sister cells does appear to me as a distinct behavior from the other intercalations shown, but i think the term "rotation" does not capture it”). With this dynamic imaging approach, we were able to dissect the influence of the PCP pathway on the distinct steps generating single column formation.

Finally, doing the single color clone in the PCP perturbations is also not very satisfying, for the same reason above, as it is difficult to understand which columns are "complex stacks" and which are "disorganized”.

We performed multiple-color clonal analysis in the PCP perturbations (Figure 4) as the reviewer requested in his/her previous comments. We agree that it is difficult to distinguish complex (multi) and disorganized columns in the PCP perturbations; hence, in the text of the revised version, we highlighted the morphologies of single-row clones (subsection “Inhibiting PCP signaling disrupts oriented cell division, but not cell pivot behavior” paragraph three).

I would disagree with their statement in their Response to Reviewers: "As mediolateral intercalation is well known to be important in cartilage development[…] we chose instead to focus[…]" I do not think this mediolateral intercalation is well-established, as the major evidence is from limited zebrafish Meckel's cartilage studies. The data in this paper still leaves me wondering: does the intercalation actually occur in this system?

Mediolateral intercalation has been demonstrated during limb cartilage growth in higher vertebrates as well. In mouse, static imaging of clones with lacZ staining strongly suggests that chondrocytes undergo an intercalation process (1). Since the labeled patches displayed more than one cell diameter in width, it is consistent with a model in which multi column formation is likely driven by mediolateral intercalation. Furthermore, it has been shown that this step is regulated by the function of proteins attached to the cell surface via glycosylphosphatidylinositol (GPI) linkages. In chick, live imaging and quantitative analysis has clearly demonstrated that mediolateral intercalation occurs in limb cartilage (2). In our view, the next challenge is to combine clonal analysis, live imaging and quantitative approaches to understand how clonal heterogeneity drives complex clone structure. As we stated above, such a study needs both new reagents and tools, and is outside the scope of the current work, but the toolkit we develop here builds a critical foundation for future mechanistic studies of chondrocyte clones with complex morphologies.

Reviewer #3:This revised manuscript is generally improved over the original, as efforts have been made to quantify the observed results.However, concerns do remain. The quantification of cytoplasmic FP intensity does allow the authors say a bit more about the data they obtained, but this is not really the correct fix. As per my original review, the absence of a membrane label is still problematic as it impedes a clear understanding of what is really going on with these cells. Given the complexity of the authors' claims, this remains a considerable issue.

We agree that a cell membrane marker would provide a more accurate means for cell surface segmentation and further cell-cell contact analysis. Indeed, we tried one membrane marker membrane RFP. However, the expression level of this marker is very weak and requires high laser power for excitation, which consequently causes photobleaching of live cartilage. To circumvent this problem, we instead have used Ncad-GFP, since it also labels cell membrane (Figure 6). Therefore, we measured fluorescence intensity changes of this fusion protein across both sisters cell during their rearrangement, as we did for cytoplasmic mCherry. The results confirm that sister cells are physically coupled with each other in wild-type and Fzd7-ΔPDB expressing tissues, but not in Fzd7 expressing ones (subsection “Influence of PCP signaling on cell pivot behavior involves junctional N-Cadherin” third paragraph) (Figure 6—figure supplement 3)

In addition, quantification of raw pixel intensity is to a useful metric of N-cad levels for Figure 6. Typically, this value would be normalized in some way (to a membrane label, or to total signal).

We normalized the signal of junctional Ncad signal to total Ncad signal (Figure 6 legend) (Figure 6).

**References:**

1. Ahrens M, Li Y, Jiang H, Dudley AT. Convergent extension movements ingrowth plate chondrocytes require gpi-anchored cell surface proteins. *Development* 136, 3463-74 (2009).

2. Li, Y. Trivedi, V. Truong, T. Koos, D. Lansford, R. Chuong, C. Warburton, D. Moats, R. Fraser, S. Dynamic imaging of the growth plate cartilage reveals multiple contributors to skeletal morphogenesis. *Nat Commun* 6, 6798 (2015).

3. Axelrod, J. D., Miller, J. R., Shulman, J. M., Moon, R. T. and Perrimon, N. Differential recruitment of Dishevelled provides signaling specificity in the planar cell polarity and Wingless signaling pathways. *Genes Dev.* 12,2610 -2622 (1998).

4. Krasnow, R. E. and Adler, P. N. A single frizzled protein has a dual function in tissue polarity. *Development* 120,1883 -1893 (1994).

5. Rothbacher, U., Laurent, M. N., Deardorff, M. A., Klein, P. S., Cho, K. W. and Fraser, S. E. Dishevelled phosphorylation, subcellular localization and multimerization regulate its role in early embryogenesis. *EMBO J.* 19,1010 -1022 (2000).

6. Li,Y, Dudley, A. Noncanonical frizzled signaling regulates cell polarity of growth plate chondrocytes. *Development* 136, 1083-92 (2009).

7. Ahrens MJ, Romereim S, Dudley, A. A re-evaluation of two key reagents for in vivo studies of Wnt signaling. Dev Dyn. 240(9):2060-8 (2011)

8. Oberlender SA, Tuan RS. Expression and functional involvement of N-cadherin in embryonic limb chondrogenesis. *Development* 120, 177-87 (1994).

9. Kintner C. Regulation of embryonic cell adhesion by the cadherin cytoplasmic domain. Cell. 69(2):225-36 (1992).